# Laplacian Representations for Decision-Time Planning

**Dikshant Shehmar** [1 2]   **Matthew Schlegel** [3]   **Matthew E. Taylor** [1 2 4]   **Marlos C. Machado** [1 2 4]

## Abstract

Planning with a learned model remains a key challenge in model-based reinforcement learning (RL) due to the compounding error problem. In decision-time planning, state representations are critical as they must support local cost computation while preserving long-horizon temporal structure. In this paper, we show that the Laplacian representation provides an effective latent space for planning by capturing state-space distances at multiple time scales. The Laplacian representation preserves meaningful distances and naturally decomposes long-horizon problems into subgoals, thus mitigating the compounding errors that arise over long prediction horizons. Building on these properties, we introduce ALPS, a hierarchical planning algorithm, and demonstrate that it outperforms commonly used model-free baselines on a selection of offline goal-conditioned RL tasks from OGBench.

## 1. Introduction

Compared to model-free RL, model-based RL offers, among other benefits, the potential for improved sample efficiency (Janner et al., 2019; Wang et al., 2019), better generalization (Deisenroth & Rasmussen, 2011; Yu et al., 2020), and faster adaptation (Zhang et al., 2020; Wan et al., 2022). These gains stem from the agent's ability to use a model to reason about consequences before acting, a process known as planning (Sutton, 1991). Nevertheless, when function approximation is required, planning with a *learned model* remains a central challenge, due to both how states are represented in latent space and generalize, and to the well-known compounding errors problem over long sequences of predictions (Talvitie, 2014; 2017; Clavera et al., 2018).

Decision-time planning algorithms such as MPC (García et al., 1989) and MCTS (Kocsis & Szepesvári, 2006) use the model to choose actions based on simulated future trajectories and predicted outcomes. In this setting, the effects of compounding errors over long horizons due to imperfect models become obvious when imagined trajectories diverge from reality (Gu et al., 2016; Lambert et al., 2022). Hierarchical planning can overcome this limitation by decoupling the agent's low-level actions from its long-horizon objectives (Koul et al., 2024), but requires a latent space in which to plan at long time scales. State representation is therefore critical: nearby states must be close in latent space, while long-term distances must also be preserved to support planning toward a desired goal.

The Laplacian representation (Mahadevan, 2005; Mahadevan & Maggioni, 2007) embeds states into a latent space defined by the eigenvectors of the graph Laplacian induced by the environment's dynamics. It captures the environment's temporal structure and connectivity, with eigenvectors ordered by time scale: early eigenvectors encode global structure (e.g., rooms or regions), while later ones capture increasingly local distinctions (Machado et al., 2017; 2023; Jinnai et al., 2019; 2020). Because this representation reflects reachability, it naturally partitions the environment into well-connected regions (Shi & Malik, 2000). Moreover, Euclidean distance in the Laplacian space approximates how easily one state can be reached from another by following the environment's dynamics (Lovász et al., 1993), yielding a geometry well suited for planning. As a result, in this paper, we show how the Laplacian representation supports both subgoal discovery and planning in a unified metric space.

This paper tackles decision-time planning with a learned model in environments requiring function approximation. We show that the Laplacian representation provides an effective latent space for hierarchical planning in long-horizon tasks, as it intrinsically captures multiple time scales and naturally decomposes tasks into subgoals. We instantiate these ideas in a novel hierarchical decision-time planning algorithm, Augmented Laplacian Planning with Subgoals (ALPS), where the Laplacian representation supports both subgoal identification and distance estimation. Empirically, ALPS outperforms the commonly used model-free RL baselines on a suite of goal-conditioned tasks from OGBench (Park et al., 2025).

---

[1]Dept. of Computing Science, University of Alberta, Canada [2]Alberta Machine Intelligence Institute (Amii) [3]Schulich School of Engineering, Dept. of Electrical and Software Engineering, University of Calgary, Canada [4]Canada CIFAR AI Chair, Amii. Correspondence to: Dikshant Shehmar <ddikshan@ualberta.ca>.

*Proceedings of the 43rd International Conference on Machine Learning*, Seoul, South Korea. PMLR 306, 2026. Copyright 2026 by the author(s).

# 2. Preliminaries

In this paper, we use lowercase symbols (e.g., $r$) for functions and values of random variables, uppercase symbols (e.g., $C$) for constants and random variables, calligraphic font (e.g., $\mathcal{S}$) for sets, bold lowercase symbols (e.g., $\mathbf{e}$) for vectors, and bold uppercase symbols (e.g., $\mathbf{L}$) for matrices. We write the $i$-th entry of a vector $\mathbf{v}$ as $\mathbf{v}(i)$.

## 2.1. Problem setting

We consider the offline Goal-Conditioned Reinforcement Learning (GCRL) setting (Kaelbling, 1993; Liu et al., 2022). The agent-environment interaction is modeled as a goal-augmented Markov decision process (GA-MDP), $\mathcal{M} = \langle \mathcal{S}, \mathcal{A}, \mathbf{P}, \mu, \gamma, \mathcal{G} \rangle$ where $\mathcal{S}$ and $\mathcal{A}$ denote the state and action spaces respectively, $\mathbf{P} : \mathcal{S} \times \mathcal{A} \to \Delta(\mathcal{S})$ denotes the state transition dynamics (where $\mathbf{P}(s'|s,a)$ is the probability of transitioning from $s$ to $s'$ when action $a$ is taken), $\mu \in \Delta(\mathcal{S})$ is the initial state distribution, $\gamma \in [0, 1]$ is the discount factor, and the goal space $\mathcal{G} \subseteq \mathcal{S}$ is introduced, where any valid state can serve as a goal (Schaul et al., 2015; Andrychowicz et al., 2017). A dataset $\mathcal{D}$ is provided consisting of trajectories $\tau^{(n)} = (s_0^{(n)}, a_0^{(n)}, s_1^{(n)}, \ldots, s_T^{(n)})$ collected from the environment beforehand through a preset behavior policy. The agent learns a goal-conditioned policy $\pi : \mathcal{S} \times \mathcal{G} \to \Delta(\mathcal{A})$ that, for each goal $g \in \mathcal{G}$, maximizes:

$$\mathbb{E}_\pi \left[ \sum_{t=0}^{T} \gamma^t r_g(S_t) \right],$$

where $r_g(s)$ is a sparse reward function (e.g., an indicator $\mathbb{I}[s = g]$), and $T \in \mathbb{N}$ denotes the episode length. However, as the horizon grows, learning a single flat policy becomes increasingly difficult as the sparse reward signal provides no learning signal until the goal is reached, making credit assignment over long action sequences challenging. Hierarchical decomposition alleviates this difficulty by introducing temporal abstraction, enabling a high-level policy to set intermediate subgoals that a low-level policy executes over shorter horizons (Nachum et al., 2018; Levy et al., 2019).

## 2.2. Laplacian representation in RL

The Laplacian framework (Mahadevan, 2005; Mahadevan & Maggioni, 2007) proposes a state representation that leverages spectral analysis to reflect global geometries of a Markov decision process (MDP). The states and transitions of an MDP are re-interpreted as nodes and edges in a weighted graph $G = (\mathcal{S}, \mathcal{E})$, where $(i, j) \in \mathcal{E}$ if the agent can observe the transition $s_i \to s_j$ in a single step; edge weights are determined by the transition matrix $\mathbf{P}_\pi$ induced by the policy $\pi$ and the environment dynamics. The graph Laplacian $\mathbf{L}$ is defined with respect to a policy $\pi$ as

$$\mathbf{L} = \mathbf{I} - f(\mathbf{P}_\pi),$$

where $\mathbf{I}$ is the identity matrix, and $f$ is a function that preserves the spectral structure of $\mathbf{P}_\pi$, commonly $f(\mathbf{P}_\pi) = \frac{1}{2}(\mathbf{P}_\pi + \mathbf{P}_\pi^\top)$ (Wu et al., 2019). If $f$ is a symmetric function, then $\mathbf{L}$ can be eigendecomposed as $\mathbf{L} = \mathbf{E}\mathbf{\Lambda}\mathbf{E}^\top$, where $\mathbf{E} = [\mathbf{u}_0, \mathbf{u}_1, \ldots, \mathbf{u}_{|\mathcal{S}|-1}]$ have eigenvectors as columns, and $\mathbf{\Lambda} = \operatorname{diag}(\lambda_0, \lambda_1, \ldots, \lambda_{|\mathcal{S}|-1})$ contains the corresponding eigenvalues.

The *Laplacian representation* is a state representation mapping $\phi : \mathcal{S} \to \mathbb{R}^D$ ($0 < D \le |\mathcal{S}|$) defined by the eigenvectors of $\mathbf{L}$ corresponding to the smallest $D$ non-zero eigenvalues (Mahadevan, 2005; Gomez et al., 2023).[1] The first eigenvector, $\mathbf{u}_0$, is constant and uninformative, and thus is discarded (Lovász et al., 1993). The representation for state $s$ is given by $\phi(s) = [\mathbf{u}_1(s), \ldots, \mathbf{u}_D(s)]^\top$, where $\mathbf{u}_i(s)$ denotes the $s$-th component of the eigenvector $\mathbf{u}_i$.

Recently, several methods have been proposed to learn the Laplacian representation from samples, thereby overcoming the high computational barrier ($O(|\mathcal{S}|^3)$) of performing an eigendecomposition of the graph Laplacian (Wu et al., 2019; Wang et al., 2021; Gomez et al., 2023). We use Augmented Lagrangian Laplacian Objective (ALLO; Gomez et al., 2023), an objective to learn the eigenvectors and eigenvalues of the graph Laplacian:

$$\max_{\boldsymbol{\beta}} \min_{\mathbf{u}} \sum_{i=1}^{D} \langle \mathbf{u}_i, \mathbf{L}\mathbf{u}_i \rangle + \sum_{j=1}^{D} \sum_{k=1}^{j} \beta_{jk} \left( \langle \mathbf{u}_j, [\![\mathbf{u}_k]\!] \rangle - \delta_{jk} \right)$$
$$+ B \sum_{j=1}^{D} \sum_{k=1}^{j} \left( \langle \mathbf{u}_j, [\![\mathbf{u}_k]\!] \rangle - \delta_{jk} \right)^2, \quad (1)$$

where $\mathbf{u}_i \in \mathbb{R}^{|\mathcal{S}|}$ denotes the $i$-th learned eigenvector, $(\beta_{jk})_{1 \le k \le j \le D}$ are the corresponding dual variables that enforce orthonormality, $\lambda_i = -\beta_{ii}/2$ are the eigenvalues, $\delta_{jk}$ is the Kronecker delta, $[\![\cdot]\!]$ denotes the stop-gradient operator, and $B$ is the barrier coefficient.[2]

## 2.3. Planning with a learned model

Model Predictive Control (MPC) is a common approach for planning in continuous action spaces. In MPC, a model is used to select the greedy action from a $k$-step lookahead search with respect to a cost function; the greedy action is then executed, and the process is repeated using the new state. The Cross-Entropy Method (CEM; Rubinstein, 1997) is a popular MPC-based planner that has been successful in model-based RL settings (Finn & Levine, 2017; Chua et al., 2018; Hafner et al., 2019; Pinneri et al., 2021; Gürtler & Martius, 2025). CEM samples action sequences using

---

[1] We define the Laplacian representation in the discrete setting for simplicity, but a principled analogous version exists for continuous state spaces (Gomez et al., 2023).

[2] Gomez et al. (2023) shows that ALLO is insensitive to the value of the barrier coefficient, which has also been our experience.

a diagonal Gaussian, $\mathbf{a}_{t:t+H} \sim \mathcal{N}(\boldsymbol{\mu}_{t:t+H}, \mathrm{diag}(\boldsymbol{\sigma}_{t:t+H}^2))$, and rolls each one out using a forward model. Based on the top-$N_e$ trajectories with the lowest cost to the goal, the mean $\boldsymbol{\mu}$ and variance $\boldsymbol{\sigma}^2$ are updated accordingly. This iterative process reliably converges to a near-optimal action sequence, but only when the planning horizon remains within the model's accuracy window (Chua et al., 2018; Feinberg et al., 2018) — beyond this, compounding model errors make long-horizon plans unreliable.

Hierarchical Reinforcement Learning (HRL) bridges this gap by decomposing the long-horizon problem into a sequence of smaller, manageable subproblems (Barto & Mahadevan, 2003; Klissarov et al., 2025), allowing each subproblem to be solved within the model's reliable planning range. Plannable Continuous Latent States (PcLast; Koul et al., 2024) is a hierarchical decision-time planning algorithm that identifies subgoals via $k$-means clustering (Lloyd, 1982) in a learned latent space, and uses CEM to plan between them. The latent space is obtained via a contrastive learning objective structured so that distances reflect random-walk reachability, making it suitable for low-level cost estimation (see Appendix B.1 for more details).

ALPS is heavily inspired by PcLast, but instead uses the Laplacian representation as its high-level latent space to capture environment geometry directly from the raw state space, and includes a behavior prior to bias CEM optimization.

## 3. The Laplacian for Long-Horizon Planning

In this section, we motivate the Laplacian representation as an effective latent space for both subgoal generation and trajectory optimization. The representation induces a metric space encoding the environment's temporal structure and connectivity at multiple time scales. Furthermore, eigenvectors of the graph Laplacian form a natural basis for clustering methods well-suited for subgoal identification (Şimşek et al., 2005; Von Luxburg, 2007). Finally, as discussed in Section 2.2, recent breakthroughs in learning the Laplacian representation from data (Wu et al., 2019; Wang et al., 2021; Gomez et al., 2023) unlock the aforementioned benefits in problems with large state and action spaces.

The distance metric induced by the Laplacian representation has a well known connection to commute time distance (CTD) of the underlying graph (Klein & Randić, 1993).[3] CTD, denoted as $c(u,v)$, is defined as the expected number of steps to travel from node $u$ to $v$ and back to $u$ following a random walk (Lovász et al., 1993), and can be computed directly from the eigenvectors and eigenvalues

---

[3]The resistance distance, $r_d(u,v) = \frac{c(u,v)}{\mathrm{vol}(G)}$ (Klein & Randić, 1993), can equivalently be used.

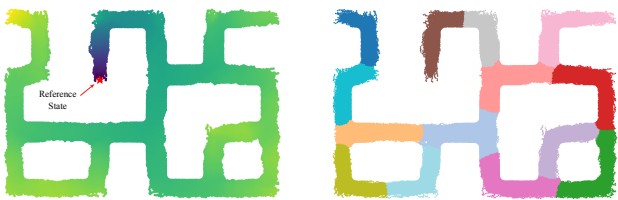

*Figure 1.* Visualization of the scaled Laplacian ($\psi$) space properties in the *pointmaze-large* environment from OGBench. (**left**) Heatmap of $c(s^\star, s_i)$ distance from a reference state ($s^\star$ denoted by $\star$ in the figure) to each state in the dataset (darker colors indicate smaller distances). (**right**) Cluster labels assigned to each state in the dataset via clustering in $\psi$-space.

of the graph Laplacian:

$$c(u,v) = \mathrm{vol}(G) \sum_{i=1}^{|\mathcal{S}|} \left( \frac{\mathbf{u}_i(u)}{\sqrt{\lambda_i}} - \frac{\mathbf{u}_i(v)}{\sqrt{\lambda_i}} \right)^2,$$

where $\mathrm{vol}(G)$ is the volume of the graph (Xiao & Gutman, 2003). To take advantage of this relation, the scaled Laplacian representation[4] $\psi_i(s) = \phi_i(s)/\sqrt{\lambda_i}$ approximates CTD as $c(s_u, s_v) \approx \|\psi(u) - \psi(v)\|^2$. For completeness, we provide a derivation of this approximation in Appendix A.1. We refer to the scaled Laplacian space learned by ALLO as $\psi$-space. Since the $\psi$-space is isometric to the commute time distance, it serves as an effective latent space for trajectory optimization.

Spectral clustering uses the Laplacian representation as a basis for clustering (Weiss, 1999; Ng et al., 2001; Von Luxburg, 2007). Intuitively, from the perspective of CTD, clustering using the Laplacian representation as a basis partitions the environment at its bottlenecks and groups well-connected regions (Shi & Malik, 2000). A random walk on the graph is unlikely to travel between different clusters, and in the Laplacian representation this is reflected in a longer distance between points from different clusters. Using the scaled Laplacian representation accentuates the relationships between points compared to the non-scaled version, pulling similar points closer together and pushing dissimilar points further apart (Qiu & Hancock, 2007). Motivated by these properties, we use the scaled Laplacian representation for subgoal identification via $k$-means clustering and perform high-level planning in $\psi$-space.

So far, we have motivated the scaled Laplacian representation as an ideal latent space for hierarchical planning by connecting it to CTD and spectral clustering. In Figure 1, we visualize the Laplacian representation learned with ALLO in the *large pointmaze* environment from OGBench (see Section 5 for more details). Note that (1) the approximation

---

[4]We call this the scaled Laplacian representation, but it has several names throughout the literature, such as reachability-aware Laplacian representation (Wang et al., 2023a), and commute-embedding (Qiu & Hancock, 2007).

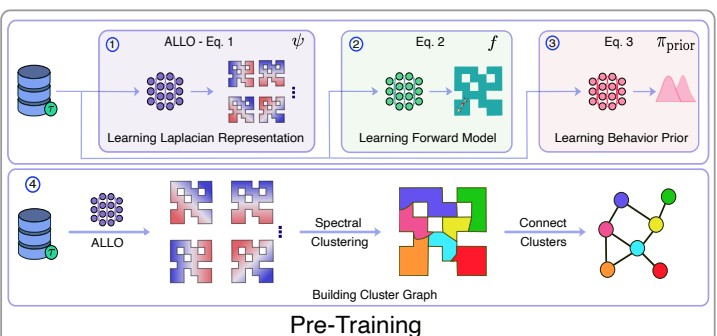 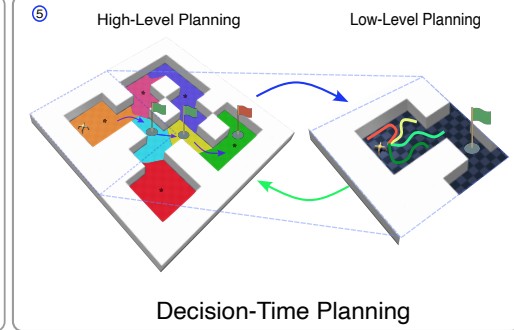

*Figure 2.* **ALPS at the pre-training and decision-time planning phases.** In pre-training, ALPS (1) learns the Laplacian representation using the Augmented Lagrangian Laplacian Objective (ALLO), (2) learns a one-step forward model on-top of the original state space $\mathcal{S}$, (3) learns the behavior prior, $\pi_{\text{prior}}$, using the scaled Laplacian representation, and (4) clusters the dataset using $k$-means in the scaled Laplacian space to generate the cluster graph. In planning, (5) ALPS takes the current state from the environment and uses the high-level planner to determine the next subgoal, and then determines the next action towards this subgoal using the low-level planner.

of CTD shows distances that are temporally consistent, and (2) the tight clusters that respect the environment dynamics whose centers can be used as subgoals. These characteristics of the scaled Laplacian representation provide a strong foundation for hierarchical decision-time planning.

# 4. Augmented Laplacian Planning with Subgoals

Augmented Laplacian Planning with Subgoals (ALPS) is a hierarchical planning algorithm that leverages the Laplacian representation to efficiently plan in continuous state and action spaces. ALPS leverages an offline dataset to learn: (1) the Laplacian representation, (2) a forward model, and (3) a behavior prior. Clustering is then performed over the scaled Laplacian representation as a basis, and the clusters centers are used as subgoals. When given a goal state, a high-level plan over subgoals is constructed using Dijkstra algorithm and this plan is executed using CEM with a behavior prior guiding its search. ALPS is outlined in Algorithm 1 with its components detailed in Fig. 2.

**Learning the Laplacian representation**: We learn the Laplacian representation, $\phi$, using the ALLO objective (Eq. 1). The ALLO objective is minimized using stochastic gradient descent by sampling transition pairs $(S_t, S_{t+\Delta})$, where $\Delta \sim \text{Geom}(1-\gamma_s)$ is the geometric distribution. The scaled Laplacian representation, $\psi$, is then obtained by scaling the individual eigenvectors with their corresponding eigenvalues obtained from dual variables:

$$\psi_i = \frac{\phi_i}{\sqrt{\lambda_i}} = \frac{\sqrt{2}\phi_i}{-\sqrt{\beta_{ii}}}.$$

**Learning a forward model**: The low-level planner uses a one-step model in the original state space $f : \mathcal{S} \times \mathcal{A} \to \mathcal{S}$. To reduce the accumulation of errors known to occur in one-step forward models, we use a multi-step auto-regressive

objective over the horizon $H_{\text{f}}$ (Talvitie, 2014; 2017):

$$\mathbb{E}_{(S_t, A_{t:t+H_{\text{f}}-1}, S_{t+1:t+H_{\text{f}}})\sim\mathcal{D}}\left[\frac{1}{H_{\text{f}}}\sum_{\tau=1}^{H_{\text{f}}}\|\hat{S}_{t+\tau} - S_{t+\tau}\|_2^2\right],$$
(2)

where $\hat{S}_{t+1} = f(S_t, A_t)$ and $\hat{S}_{t+\tau} = f(\hat{S}_{t+\tau-1}, A_{t+\tau-1})$ for $\tau > 1$. Auto-regressive training allows gradients to flow back through time, incentivizing the model to learn transition dynamics that remain stable over $H_{\text{f}}$.

**Learning a behavior prior**: Usually, CEM samples action sequences from an unconditional Gaussian distribution, but this strategy is inefficient for high-dimensional action spaces (Bharadhwaj et al., 2020). To accelerate the convergence of CEM, we use a behavior prior to generate candidate action sequences (Hansen et al., 2024). The behavior prior, $\pi_{\text{prior}}(S_t, \psi(S_t), \psi(S_{t+k}))$ is a deterministic goal-conditioned policy. We learn $\pi_{\text{prior}}$ by minimizing

$$\mathbb{E}_{(S_t, A_t, S_{t+k})\sim\mathcal{D}}[\|\pi_{\text{prior}}(S_t, \psi(S_t), \psi(S_{t+k}))-A_t\|_2^2], \quad (3)$$

where $k \sim U(1, K_{\text{max}})$, with $K_{\text{max}}$ as a hyperparameter. This behavior cloning objective encourages goal-directed behavior by assuming all trajectories in the dataset are generated by a goal-seeking policy trying to reach the goal state $S_{t+k}$ from $S_t$. The behavior prior is then used to predict an action sequence $\mathbf{a}_{t:t+H-1}$ from a state $S_t$ using the forward model over the planner horizon $H$.

**Building a cluster graph**: The space is partitioned into $C$ regions $\{c_i\}_{i=1}^C$ using $k$-means clustering in the $\psi$-space. The cluster centers act as the vertices of a graph $G_c$, with the edges defined by the dataset $\mathcal{D}$: an edge $(i,j)$ exists if we observe transitions from states that belong to cluster $i$ to states from cluster $j$ (or the converse). Like PcLast, we prune infrequent inter-cluster transitions via nucleus sampling (Holtzman et al., 2020), retaining only top-$p\%$ most frequent neighbors of each cluster to avoid unreachable subgoals.

**Algorithm 1** ALPS

> **Training Input**: Dataset $\mathcal{D}$, number of eigenvectors $D$, number of clusters $C$
> === Pre-Training Phase ===
> $\psi \leftarrow \text{TrainALLO}(\mathcal{D})$        $\triangleright \psi : \mathcal{S} \to \mathbb{R}^d$
> $f \leftarrow \text{TrainDynamics}(\mathcal{D})$      $\triangleright f : \mathcal{S} \times \mathcal{A} \to \mathcal{S}$
> $\pi_{\text{prior}} \leftarrow \text{TrainBehaviorPrior}(\mathcal{D}, \psi)$   $\triangleright \pi_{\text{prior}} : \mathcal{S} \times \psi \times \psi \to \mathcal{A}$
> === Get Cluster Graph ===
> labels $\leftarrow \text{KMeans}(\psi, \mathcal{D}, C)$
> $G_c \leftarrow \text{BuildClusterGraph}(\mathcal{D}, \text{labels})$
> === Decision-Time Planning ===
> **Testing Input**: $s_{\text{start}}, s_{\text{goal}}$      $\triangleright$ get start and goal state
> $z_s \leftarrow \psi(s_{\text{start}}), z_g \leftarrow \psi(s_{\text{goal}})$
> $c_s \leftarrow \text{getCluster}(z_s), c_g \leftarrow \text{getCluster}(z_g)$   $\triangleright$ get cluster label
> $\mathcal{P}_{\mathcal{G}} \leftarrow \text{Dijkstra}(G_c, c_s, c_g)$      $\triangleright$ get high-level plan
> $i \leftarrow 1$       $\triangleright$ index of next target cluster
> **while** $\|\psi(s) - z_g\| > \epsilon$ **do**
>     $c_{\text{curr}} \leftarrow \text{getCluster}(\psi(s))$
>     **if** $c_{\text{curr}} \notin \mathcal{P}_{\mathcal{G}}$ **then**
>       $\mathcal{P}_{\mathcal{G}} \leftarrow \text{Dijkstra}(G_c, c_{\text{curr}}, c_g)$   $\triangleright$ replan if drifted from $\mathcal{P}_{\mathcal{G}}$
>       $i \leftarrow 1$
>     **else if** $c_{\text{curr}} = \mathcal{P}_{\mathcal{G}}[i]$ **then**
>       $i \leftarrow \min(i+1, |\mathcal{P}_{\mathcal{G}}| - 1)$    $\triangleright$ advance if reached target
>     **end if**
>     $z_{\text{sub}} \leftarrow \begin{cases} z_g & \text{if } \mathcal{P}_{\mathcal{G}}[i] = c_g \\ \text{centroid}(\mathcal{P}_{\mathcal{G}}[i]) & \text{otherwise} \end{cases}$
>     $\mathbf{a} \leftarrow \text{CEM}(s, z_{\text{sub}}, \psi, f, \pi_{\text{prior}})$
>     $s \leftarrow \text{env.step}(\mathbf{a}[0])$
> **end while**

**Decision-time planning**: At the beginning of an episode, the planner receives the start state, $s_{\text{start}}$, and the goal state, $s_{\text{goal}}$. It then identifies their respective clusters, $c_s$ and $c_g$, in the cluster graph $G_c$. Dijkstra's algorithm (Dijkstra, 1959) finds the shortest path $\mathcal{P}_{\mathcal{G}}$ on the cluster graph between the clusters $c_s$ and $c_g$. $\mathcal{P}_{\mathcal{G}}$ acts as a high-level plan for the agent, providing a path in the scaled Laplacian $\psi$-space.

At each step of the episode, the high-level planner provides the next target cluster center, $z_{\text{sub}}$, from $\mathcal{P}_{\mathcal{G}}$. The low-level planner then uses CEM to find an action that moves the agent toward $z_{\text{sub}}$. First, a mean action sequence, $\mathbf{a}_{t:t+H-1} = (A_t, A_{t+1}, \ldots, A_{t+H-1})$, is generated by rolling out the behavior prior $\pi_{\text{prior}}$ autoregressively through the learned forward model $f$: at each step $k$, the prior proposes an action given the current predicted state, and $f$ predicts the next state $\hat{S}_{t+k+1} = f(\hat{S}_{t+k}, a_{t+k})$. Then, $N_s$ candidate action sequences are generated by adding temporally-correlated Gaussian noise (Wang & Ba, 2020) around this mean sequence. Each candidate is rolled out through the forward model to produce a trajectory, which is evaluated using the cost function:

$$J^m = \sum_{t=1}^{H} \left( \|\psi(\hat{S}_t^{\,m}) - z_{\text{sub}}\|_2^2 + \lambda \|A_t^m\|_2^2 \right), \quad (4)$$

where the first term minimizes the distance in Laplacian space and the second term penalizes large actions with $\lambda$

as a hyperparameter. Note that since we perform spectral clustering in $\psi$-space, the distance between $\psi(\hat{S}_t)$ and $z_{\text{sub}}$ approximates the CTD between corresponding states. We penalize extreme actions to prevent abrupt behavior. The CEM algorithm is outlined in Algorithm 2. The top-$N_e$ lowest cost trajectories are then used to update the action sampling distribution, and this process is repeated over $N_{\text{iter}}$ iterations. The agent then executes the first action of the best action sequence. If the agent deviates from the precomputed plan, the high-level planner recomputes a plan from the current cluster to the goal state.

Once the agent enters the current target cluster, the next cluster center in $\mathcal{P}_{\mathcal{G}}$ becomes the target cluster. This process repeats until the agent reaches the goal cluster, when CEM uses the goal state as the final target. The pseudocode of the CEM planner is outlined in Appendix D.

## 5. Experiments

We now empirically validate the utility of the Laplacian representation as a latent space for decision-time planning with ALPS. First, given that ALPS shares many of the same components as PcLast, we directly compare these algorithms on Maze2D—Point Mass tasks, where PcLast was originally evaluated. These environments pose difficult navigation conditions for long-horizon planning and state-space partitioning from images, as they require the agent to travel far to reach states that appear close in Euclidean space.

To demonstrate ALPS's scalability and potential, we evaluate it on the locomotion and manipulation tasks from Offline-Goal Conditioned RL Benchmark suite (OGBench; Park et al., 2025), a complex robotic benchmark for offline GCRL with large state and action spaces. The locomotion tasks are difficult not only because navigating from start to goal requires navigating a maze, but because the underlying locomotion is itself challenging. The manipulation tasks are designed to test the agent's object manipulation, sequential generalization, and combinatorial generalization abilities. Up-to-now, model-free baselines have led performance across all OGBench tasks. The code is available at https://github.com/machado-research/ALPS.[5]

### 5.1. Experimental domains

In all environments, performance is measured by the average success rate across five pre-defined state-goal pairs. In each evaluation, a goal $g$ is given to the agent, and the episode immediately terminates when the agent reaches the goal. Unless otherwise specified, all results report mean and standard deviation over 8 seeds for state-based tasks and 4 seeds for pixel-based tasks. Hyperparameter details are in Appendix F.

---

[5]Project Page: https://dikshuy.github.io/ALPS/

**Maze2D—PointMass** (Koul et al., 2024): Each maze is a unit square with varied wall configurations (see Figure 4 in Appendix E.1). The agent controls a point mass with actions corresponding to the coordinate space change $(\Delta x, \Delta y)$ bounded by the range $[-0.2, 0.2]$ for each action. Observations are defined as a single-channel $(100 \times 100)$ image encoding the current position of the agent and no other environmental information. A Gaussian blur $(\sigma = 1.0)$ is applied to the agent's coordinate position, and the resulting image is normalized to $[0, 1]$. An offline dataset of $500K$ transitions is generated using a uniform random policy. We follow PcLast's empirical design where the agent must navigate from a starting position to within 0.03 units of a known target position within 30 actions.

**Locomotion and manipulation tasks from OGBench** (Park et al., 2025): We use three locomotion tasks from OGBench: *pointmaze*, *antmaze*, and *humanoidmaze*, requiring control of a 2-DoF *ball*, 8-DoF *ant*, and 21-DoF *humanoid* body, respectively. We consider both state-based and pixel-based variants. In state-based variants, the agent has access to the full low-dimensional state, including its $x$-$y$ position; in pixel-based variants, it receives only $64 \times 64 \times 3$ third-person images, with the floor colored to enable location inference without recurrent networks. The dataset types are as follows: (i) *navigate*, collected by a noisy expert repeatedly reaching random sampled goals; (ii) *stitch*, collected through shorter goal-reaching trajectories testing stitching ability; and (iii) *explore*, collected with high action noise, testing navigation from extremely low-quality but high-coverage data.

We consider two robotic manipulation tasks: *Cube* and *Scene*, designed to test object manipulation, sequential generalization, and combinatorial generalization. Both use a 6-DoF UR5e arm with a Robotiq 2F-85 gripper controlled via a 5-D end-effector action space. *Cube* tasks involve pick-and-place manipulation of blocks into a desired configuration; *Scene* tasks require pressing a button to toggle lock states, picking up a cube, and placing it in a drawer. Datasets are collected by scripted non-Markovian policies with temporally correlated noise (Park et al., 2025).

An episode terminates as soon as the agent reaches the proximity of the goal location that defines success, or when the episode ends, which varies across environments. For locomotion tasks, success is determined by proximity to the goal location, not joint positions (Park et al., 2023). For manipulation tasks, success is based solely on object configurations. We follow the OGBench evaluation protocol, averaging performance over 750 rollouts (3 evaluation epochs × 5 test-time goals × 50 rollouts). See Appendix E.2 for further details about environments and evaluation setup.

We compare to the baseline algorithms selected by Park et al. (2025). Goal-conditioned behavior cloning (GCBC; Lynch et al., 2020; Ghosh et al., 2021) performs goal-conditioned behavior cloning by sampling a future state from the same trajectory as the goal. Goal-conditioned implicit {V, Q}-learning (GCIVL; Kostrikov et al., 2022), (GCIQL; Park et al., 2024a) are goal-conditioned variants of implicit Q-Learning (IQL; Kostrikov et al., 2022) that fit optimal value functions ($V^*$ and $Q^*$) via expectile regression. Quasimetric RL (QRL; Wang et al., 2023b) learns a quasimetric distance function satisfying the triangle inequality to represent goal-conditioned value functions. Contrastive RL (CRL; Eysenbach et al., 2022) uses contrastive learning to learn a goal-conditioned value function. Finally, hierarchical implicit Q-learning (HIQL; Park et al., 2023) is a hierarchical model-free algorithm in which the high-level policy predicts the representation of an optimal $k$-step subgoal, and the low-level policy predicts the optimal action for this subgoal.

### 5.2. Comparison with PcLast

ALPS differs from PcLast by the latent space used for the high-level planner and for trajectory optimization, as well as by the fact that ALPS uses a behavior prior to accelerate planner convergence. To evaluate the impact of using different latent spaces, isolating the impact of the representation used, we compare the performance of PcLast to ALPS when not using a behavior prior, which we denote by ALPS[†]. We consider two scenarios, one with only the low-level planner, using a single cluster, and one with the high-level planner, using 16 clusters. The results are shown in Table 1.

*Table 1.* Success rate (%) of PcLast and ALPS[†] on the Maze2D—PointMass environments. Mean and std. deviation over 10 seeds.

| Environment | Clusters | PcLast | ALPS[†] |
|---|---|---|---|
| Hallway | 1 | $51 \pm 4$ | $94 \pm 3$ |
| | 16 | $62 \pm 4$ | $97 \pm 2$ |
| Rooms | 1 | $30 \pm 3$ | $92 \pm 3$ |
| | 16 | $57 \pm 10$ | $96 \pm 2$ |
| Spiral | 1 | $35 \pm 4$ | $91 \pm 4$ |
| | 16 | $60 \pm 6$ | $94 \pm 2$ |

Overall, ALPS[†] outperforms PcLast in both scenarios. Both algorithms perform better across all domains when using both the high- and low-level planners jointly, demonstrating that they learn a latent space useful for planning. Our distance metric's ability to disentangle the state space with respect to the dynamics can be clearly seen in the Spiral domain when the high-level planner is not included, resulting in similar performance in the low- and high-level settings with ALPS[†]. However, PcLast has a substantial performance reduction when not using the high-level planner (i.e., clusters=1). The above results demonstrate the scaled Laplacian representation's utility as both a distance metric for the low-level planner and a latent representation for subgoal identification through spectral clustering.

*Table 2.* **Success rate (%) on each of the locomotion and manipulation tasks from OGBench considered.** The results are averaged over 8 seeds (4 seeds for pixel-based tasks), and we report the standard deviation after the ± sign. ALPS outperforms the model-free GCRL algorithms with $p < 0.001$ using a Holm-Bonferroni-corrected two-sided Wilcoxon signed-rank test. Values in **bold** denote the largest mean in each row as a visual aid.

| Environment | Dataset Type | Dataset | GCBC | GCIVL | GCIQL | QRL | CRL | HIQL | ALPS |
|---|---|---|---|---|---|---|---|---|---|
| pointmaze | navigate | pointmaze-medium-navigate-v0 | $9 \pm 6$ | $63 \pm 6$ | $53 \pm 8$ | $\mathbf{82} \pm 5$ | $29 \pm 7$ | $79 \pm 5$ | $\mathbf{82} \pm 10$ |
| | | pointmaze-large-navigate-v0 | $29 \pm 6$ | $45 \pm 5$ | $34 \pm 3$ | $\mathbf{86} \pm 9$ | $39 \pm 7$ | $58 \pm 5$ | $80 \pm 8$ |
| | | pointmaze-giant-navigate-v0 | $1 \pm 2$ | $0 \pm 0$ | $0 \pm 0$ | $\mathbf{68} \pm 7$ | $27 \pm 10$ | $46 \pm 9$ | $67 \pm 11$ |
| | | pointmaze-teleport-navigate-v0 | $25 \pm 3$ | $\mathbf{45} \pm 3$ | $24 \pm 7$ | $4 \pm 4$ | $24 \pm 6$ | $18 \pm 4$ | $40 \pm 6$ |
| | stitch | pointmaze-medium-stitch-v0 | $23 \pm 18$ | $70 \pm 14$ | $21 \pm 9$ | $80 \pm 12$ | $0 \pm 1$ | $74 \pm 6$ | $\mathbf{94} \pm 6$ |
| | | pointmaze-large-stitch-v0 | $7 \pm 5$ | $12 \pm 6$ | $31 \pm 2$ | $84 \pm 15$ | $0 \pm 0$ | $13 \pm 6$ | $\mathbf{96} \pm 2$ |
| | | pointmaze-giant-stitch-v0 | $0 \pm 0$ | $0 \pm 0$ | $0 \pm 0$ | $50 \pm 8$ | $0 \pm 0$ | $0 \pm 0$ | $\mathbf{98} \pm 1$ |
| | | pointmaze-teleport-stitch-v0 | $31 \pm 9$ | $\mathbf{44} \pm 2$ | $25 \pm 3$ | $9 \pm 5$ | $4 \pm 3$ | $34 \pm 4$ | $13 \pm 4$ |
| antmaze | navigate | antmaze-medium-navigate-v0 | $29 \pm 4$ | $72 \pm 8$ | $71 \pm 4$ | $88 \pm 3$ | $95 \pm 1$ | $96 \pm 1$ | $\mathbf{97} \pm 2$ |
| | | antmaze-large-navigate-v0 | $24 \pm 2$ | $16 \pm 5$ | $34 \pm 4$ | $75 \pm 6$ | $83 \pm 4$ | $91 \pm 2$ | $\mathbf{93} \pm 5$ |
| | | antmaze-giant-navigate-v0 | $0 \pm 0$ | $0 \pm 0$ | $0 \pm 0$ | $14 \pm 3$ | $16 \pm 3$ | $65 \pm 5$ | $\mathbf{69} \pm 9$ |
| | | antmaze-teleport-navigate-v0 | $26 \pm 3$ | $39 \pm 3$ | $35 \pm 5$ | $35 \pm 5$ | $\mathbf{53} \pm 2$ | $42 \pm 3$ | $45 \pm 3$ |
| | stitch | antmaze-medium-stitch-v0 | $45 \pm 11$ | $44 \pm 6$ | $29 \pm 6$ | $59 \pm 7$ | $53 \pm 6$ | $\mathbf{94} \pm 1$ | $93 \pm 7$ |
| | | antmaze-large-stitch-v0 | $3 \pm 3$ | $18 \pm 2$ | $7 \pm 2$ | $18 \pm 2$ | $11 \pm 2$ | $67 \pm 5$ | $\mathbf{95} \pm 2$ |
| | | antmaze-giant-stitch-v0 | $0 \pm 0$ | $0 \pm 0$ | $0 \pm 0$ | $0 \pm 0$ | $0 \pm 0$ | $2 \pm 2$ | $\mathbf{92} \pm 3$ |
| | | antmaze-teleport-stitch-v0 | $31 \pm 6$ | $\mathbf{39} \pm 3$ | $17 \pm 2$ | $24 \pm 5$ | $31 \pm 4$ | $36 \pm 2$ | $35 \pm 11$ |
| | explore | antmaze-medium-explore-v0 | $2 \pm 1$ | $19 \pm 3$ | $13 \pm 2$ | $1 \pm 1$ | $3 \pm 2$ | $37 \pm 10$ | $\mathbf{100} \pm 0$ |
| | | antmaze-large-explore-v0 | $0 \pm 0$ | $10 \pm 3$ | $0 \pm 0$ | $0 \pm 0$ | $0 \pm 0$ | $4 \pm 5$ | $\mathbf{90} \pm 15$ |
| | | antmaze-teleport-explore-v0 | $2 \pm 1$ | $32 \pm 2$ | $7 \pm 3$ | $2 \pm 2$ | $20 \pm 2$ | $34 \pm 15$ | $\mathbf{48} \pm 6$ |
| humanoidmaze | navigate | humanoidmaze-medium-navigate-v0 | $8 \pm 2$ | $24 \pm 2$ | $27 \pm 2$ | $21 \pm 8$ | $60 \pm 4$ | $\mathbf{89} \pm 2$ | $\mathbf{89} \pm 5$ |
| | | humanoidmaze-large-navigate-v0 | $1 \pm 0$ | $2 \pm 1$ | $2 \pm 1$ | $5 \pm 1$ | $24 \pm 4$ | $49 \pm 4$ | $\mathbf{56} \pm 5$ |
| | | humanoidmaze-giant-navigate-v0 | $0 \pm 0$ | $0 \pm 0$ | $0 \pm 0$ | $1 \pm 0$ | $3 \pm 2$ | $12 \pm 4$ | $\mathbf{67} \pm 11$ |
| | stitch | humanoidmaze-medium-stitch-v0 | $29 \pm 5$ | $12 \pm 2$ | $12 \pm 3$ | $18 \pm 2$ | $36 \pm 2$ | $\mathbf{88} \pm 2$ | $68 \pm 5$ |
| | | humanoidmaze-large-stitch-v0 | $6 \pm 3$ | $1 \pm 1$ | $0 \pm 0$ | $3 \pm 1$ | $4 \pm 1$ | $28 \pm 3$ | $\mathbf{39} \pm 6$ |
| | | humanoidmaze-giant-stitch-v0 | $0 \pm 0$ | $0 \pm 0$ | $0 \pm 0$ | $0 \pm 0$ | $0 \pm 0$ | $3 \pm 2$ | $\mathbf{62} \pm 6$ |
| visual-antmaze | navigate | visual-antmaze-medium-navigate-v0 | $11 \pm 2$ | $22 \pm 2$ | $11 \pm 1$ | $0 \pm 0$ | $\mathbf{94} \pm 1$ | $93 \pm 3$ | $\mathbf{94} \pm 5$ |
| | | visual-antmaze-large-navigate-v0 | $4 \pm 0$ | $5 \pm 1$ | $4 \pm 1$ | $0 \pm 0$ | $84 \pm 1$ | $53 \pm 9$ | $\mathbf{88} \pm 3$ |
| | | visual-antmaze-giant-navigate-v0 | $0 \pm 1$ | $1 \pm 1$ | $0 \pm 0$ | $0 \pm 0$ | $\mathbf{47} \pm 2$ | $6 \pm 4$ | $36 \pm 7$ |
| | | visual-antmaze-teleport-navigate-v0 | $5 \pm 1$ | $8 \pm 1$ | $6 \pm 1$ | $6 \pm 3$ | $\mathbf{48} \pm 2$ | $37 \pm 2$ | $47 \pm 3$ |
| | stitch | visual-antmaze-medium-stitch-v0 | $67 \pm 4$ | $6 \pm 2$ | $2 \pm 0$ | $0 \pm 0$ | $69 \pm 2$ | $87 \pm 2$ | $\mathbf{95} \pm 2$ |
| | | visual-antmaze-large-stitch-v0 | $24 \pm 3$ | $1 \pm 1$ | $0 \pm 0$ | $1 \pm 1$ | $11 \pm 3$ | $28 \pm 0$ | $\mathbf{90} \pm 2$ |
| | | visual-antmaze-giant-stitch-v0 | $0 \pm 0$ | $0 \pm 0$ | $0 \pm 0$ | $0 \pm 0$ | $0 \pm 0$ | $0 \pm 0$ | $\mathbf{55} \pm 6$ |
| | | visual-antmaze-teleport-stitch-v0 | $32 \pm 3$ | $1 \pm 1$ | $1 \pm 0$ | $1 \pm 2$ | $32 \pm 6$ | $\mathbf{37} \pm 2$ | $21 \pm 4$ |
| | explore | visual-antmaze-medium-explore-v0 | $0 \pm 0$ | $0 \pm 0$ | $0 \pm 0$ | $0 \pm 0$ | $0 \pm 0$ | $0 \pm 0$ | $\mathbf{78} \pm 10$ |
| | | visual-antmaze-large-explore-v0 | $0 \pm 0$ | $1 \pm 0$ | $0 \pm 0$ | $0 \pm 0$ | $1 \pm 0$ | $0 \pm 0$ | $\mathbf{26} \pm 10$ |
| | | visual-antmaze-teleport-explore-v0 | $0 \pm 0$ | $0 \pm 0$ | $0 \pm 0$ | $0 \pm 0$ | $1 \pm 0$ | $19 \pm 8$ | $\mathbf{34} \pm 4$ |
| cube | play | cube-single-play-v0 | $6 \pm 2$ | $53 \pm 4$ | $\mathbf{68} \pm 6$ | $5 \pm 1$ | $19 \pm 2$ | $15 \pm 3$ | $\mathbf{68} \pm 6$ |
| | | cube-double-play-v0 | $1 \pm 1$ | $36 \pm 5$ | $\mathbf{40} \pm 5$ | $1 \pm 0$ | $10 \pm 2$ | $6 \pm 2$ | $2 \pm 1$ |
| scene | play | scene-play-v0 | $5 \pm 1$ | $42 \pm 4$ | $\mathbf{51} \pm 4$ | $5 \pm 1$ | $19 \pm 2$ | $38 \pm 3$ | $26 \pm 3$ |

## 5.3. Scaling to large state and action spaces

We now empirically demonstrate that ALPS also scales to large state and action spaces. The results are reported in Table 2. ALPS significantly outperforms the baselines across domains according to a two-sided paired Wilcoxon signed-rank test applied to per-domain mean performance, with Holm-Bonferroni correction $p < 0.001$; see Appendix C for further details. Importantly, our results are significant because all the GCRL baselines we considered are model-free methods, due to the historical difficulty of getting model-based methods to succeed in these tasks. Furthermore, note that ALPS is more robust to the size of the maze the agent navigates, handily outperforming model-free baselines in the giant mazes. This result provides evidence that the Laplacian representation can be a useful building block for successful planning with a learned model, the central hypothesis of this thesis.

Three sets of results are particularly interesting to discuss here. First, ALPS's performance is closer to the baseline on the *pointmaze-navigate* tasks. These are the simplest tasks in OGBench, and here the behavior prior hinders

performance. When this prior is removed, ALPS achieves a success rate close to $100\%$ in the medium and large mazes, as shown in the ablation study in the next section. This is not surprising, as the introduced inductive bias is intended to help in substantially harder tasks.

The *teleport* mazes are also particularly interesting in the context of the Laplacian representation. When an agent enters a teleporter, it may reappear elsewhere in the maze or in an inescapable absorbing region from which the goal cannot be reached. Despite ALPS's strong overall performance in these environments, inspection reveals it would be unable to achieve a $100\%$ success rate. This limitation arises from the symmetries enforced by the ALLO objective: the learned Laplacian representation embeds teleporter entrances and exits as nearby states, incentivizing the agent to use the teleporter, which is suboptimal due to the risk of entering an inescapable region. We verify this empirically in Fig. 9 where all the teleport gates are getting clustered to the same cluster. Thus, if a goal is closer to the teleport gates, the agent attempts to access them, which is suboptimal due to the risk of entering an inescapable region. To the best

*Table 3.* **Success rates (%) for the six variants of ALPS compared in Section 6 on the state-based OGBench locomotion tasks.** Results averaged over 8 seeds with standard deviation reported after ±. CEM, $\pi_{\text{prior}}$, and CEM + $\pi_{\text{prior}}$ are the low-level planning only variants. Dijkstra + CEM and Dijkstra + $\pi_{\text{prior}}$ are the hierarchical planning variants with either CEM or $\pi_{\text{prior}}$ acting as the low-level planner.

| Environment | Dataset | CEM | $\pi_{\text{prior}}$ | CEM + $\pi_{\text{prior}}$ | Dijkstra + CEM | Dijkstra + $\pi_{\text{prior}}$ | ALPS |
|---|---|---|---|---|---|---|---|
| pointmaze | pointmaze-medium-navigate-v0 | $58 \pm 8$ | $19 \pm 8$ | $30 \pm 7$ | $100 \pm 0$ | $46 \pm 10$ | $82 \pm 10$ |
| | pointmaze-large-navigate-v0 | $19 \pm 11$ | $17 \pm 10$ | $21 \pm 9$ | $100 \pm 0$ | $44 \pm 17$ | $80 \pm 8$ |
| | pointmaze-medium-stitch-v0 | $40 \pm 13$ | $40 \pm 12$ | $46 \pm 11$ | $99 \pm 3$ | $82 \pm 10$ | $94 \pm 6$ |
| | pointmaze-large-stitch-v0 | $0 \pm 0$ | $12 \pm 9$ | $9 \pm 9$ | $100 \pm 1$ | $93 \pm 4$ | $96 \pm 2$ |
| antmaze | antmaze-medium-navigate-v0 | $0 \pm 0$ | $50 \pm 10$ | $57 \pm 8$ | $0 \pm 0$ | $92 \pm 5$ | $97 \pm 2$ |
| | antmaze-large-navigate-v0 | $0 \pm 0$ | $30 \pm 11$ | $29 \pm 12$ | $0 \pm 0$ | $90 \pm 4$ | $93 \pm 5$ |
| | antmaze-medium-stitch-v0 | $0 \pm 0$ | $50 \pm 13$ | $55 \pm 12$ | $0 \pm 0$ | $92 \pm 4$ | $93 \pm 7$ |
| | antmaze-large-stitch-v0 | $0 \pm 0$ | $14 \pm 8$ | $11 \pm 7$ | $0 \pm 0$ | $93 \pm 2$ | $95 \pm 2$ |
| | antmaze-medium-explore-v0 | $0 \pm 0$ | $11 \pm 6$ | $7 \pm 3$ | $47 \pm 22$ | $92 \pm 4$ | $100 \pm 0$ |
| | antmaze-large-explore-v0 | $0 \pm 0$ | $1 \pm 1$ | $0 \pm 0$ | $11 \pm 8$ | $43 \pm 9$ | $90 \pm 15$ |
| humanoidmaze | humanoidmaze-medium-navigate-v0 | $0 \pm 0$ | $29 \pm 12$ | $26 \pm 10$ | $0 \pm 0$ | $86 \pm 6$ | $89 \pm 5$ |
| | humanoidmaze-large-navigate-v0 | $0 \pm 0$ | $3 \pm 2$ | $3 \pm 2$ | $0 \pm 0$ | $58 \pm 6$ | $56 \pm 5$ |
| | humanoidmaze-medium-stitch-v0 | $0 \pm 0$ | $33 \pm 8$ | $31 \pm 7$ | $0 \pm 0$ | $64 \pm 6$ | $68 \pm 5$ |
| | humanoidmaze-large-stitch-v0 | $0 \pm 0$ | $2 \pm 2$ | $1 \pm 2$ | $0 \pm 0$ | $48 \pm 5$ | $56 \pm 5$ |

of our knowledge, these results are the first to suggest that the symmetrization function commonly used when learning Laplacian representations can be problematic, as prior work has reported success in asymmetric environments (Klissarov & Machado, 2023).

While ALPS achieves approximately 70% success rate on the *single-cube* task, performance degrades drastically on multi-cube variants. We attribute this drop to the global nature of the Laplacian representation and the compounding dimensionality of the joint state space, where each additional cube introduces 9 new dimensions. In the multi-object manipulation, task dynamics are often conditionally independent; the resting position of a distractor cube is irrelevant while the arm manipulates a target cube. Yet the Laplacian representation inherently models the global state space topology, failing to capture this spatial invariance, mapping two trajectories with identical active-cube movements but differing resting-cube positions to entirely separate, distant regions in the latent space. We believe future approaches should address this through factorized representation spaces.

## 6. Ablation Studies

Results thus far demonstrate that the Laplacian representation can be used in hierarchical decision-time planning with images and in large continuous state and action spaces. This section provides additional analysis to better understand ALPS's empirical success.

### 6.1. Components of ALPS

We perform a series of ablations to investigate how the components of ALPS influence the performance of the agent. Our aim is to disentangle the roles of CEM and the behavior prior in the low-level planner and to understand the contribution of the high-level planner to ALPS' per-

formance on the OGBench tasks. We compare ALPS with five variations, each including one or two of the following components: the high-level planner (Dijkstra+), CEM, and $\pi_{\text{prior}}$. We perform hyperparameter tuning for all ablations, reported in Table 8. A subset of results are presented in Table 3, with full results reported in Table 17. We follow the same evaluation procedure described in Section 5.

**The role of the behavior prior**: Without the behavior prior (CEM, Dijkstra + CEM), performance dramatically reduces in locomotion tasks with challenging movement dynamics like *ant* and *humanoid*. We suspect two reasons for this observation. First, this reflects the slow convergence of CEM when only using noise to generate proposal trajectories, the original motivation for including a behavior prior. Second, we believe that random actions lead to out-of-distribution transitions for the forward model. The effect of the forward model being partially responsible due to out-of-distribution transitions can be observed in the outlier performance of the *explore* dataset, which has greater state-action space coverage.

**The role of CEM**: CEM is most impactful when the expert policy used to generate the dataset is not ideal. This is supported by the reduced performance of Dijkstra + $\pi_{\text{prior}}$ in the *explore* datasets as compared to ALPS. More surprising is the favorable performance of Dijkstra + CEM over Dijkstra + $\pi_{\text{prior}}$ in the pointmaze task with the *navigate* dataset. We believe this is primarily due to the structure of the *navigate* dataset. Because the agent visits multiple goals in sequence within a single trajectory, the dataset contains both approaching and departing behaviors with respect to each goal. As a result, the behavior prior may learn a stationary policy in regions near the goal. We observed this behavior in these settings when investigating this performance.

**The role of hierarchical planning**: The low-level planner alone is unable to effectively reach the goal in the loco-

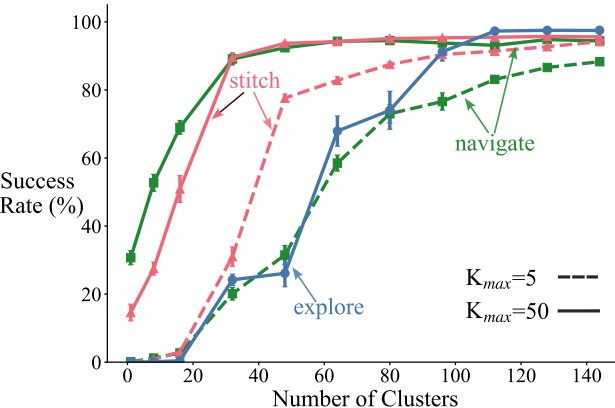

*Figure 3.* Success rate vs. number of clusters on OGBench *large* maze for *ant* task and types (*navigate*, *stitch*, *explore*). Error bars indicate standard error.

motion environments, as seen by the drop in performance across the CEM, $\pi_{\text{prior}}$, and CEM + $\pi_{\text{prior}}$ variants. When the hierarchical planner is added back (denoted by "Dijkstra+") the results generally improve across domains, and especially in domains with noisy data-generating policies (e.g., *explore* datasets). The impact of this hierarchical decomposition through an ablation on different number of clusters is expanded on below. This result demonstrates the importance of hierarchical planning for long-horizon planning in ALPS.

### 6.2. Number of clusters

We investigate how the performance of ALPS is impacted by the number of clusters used by the high-level planner and the effective planning horizon of the behavior prior, i.e., the state-to-subgoal distance at which the behavior prior can produce good plans. The results, reported in Figure 3, reveal why ALPS is so dominant on the explore datasets. As the effective horizon of the data generation policy reduces, i.e., when we artificially reduce $K_{\text{max}}$ or when the expert-data quality is reduced as in the *explore* dataset, we can simply increase the number of partitions in our high-level planner to overcome the limitation of the low-level planner. In other words, ALPS is robust to the quality of the process used to generate the expert dataset, unlike all the baselines.

### 7. Related Work

Several methods have been proposed to solve the GCRL problem. A common strategy is to condition a value function or policy on the goal (Schaul et al., 2015; Nair et al., 2018; Chane-Sane et al., 2021), but this approach struggles as the horizon grows, since sparse rewards provide no learning signal until the goal is reached, making credit assignment over long action sequences challenging (Nachum et al., 2018; Levy et al., 2019). Hierarchical approaches address this by introducing temporal abstraction, decomposing the problem into manageable subproblems where a high-level policy

sets intermediate subgoals executed by a low-level policy over shorter horizons. This can be done model-free (Levy et al., 2019; Park et al., 2023) or, like ALPS, model-based using MPC for long-term reasoning (Nasiriany et al., 2019; Nair & Finn, 2020; Pertsch et al., 2021).

Although hierarchical strategies show considerable promise for long-horizon problems, identifying subgoals remains a key challenge. Graph partitioning has been widely used to identify bottleneck states in the transition graph (McGovern & Barto, 2001; Menache et al., 2002; Şimşek et al., 2005), while others learn structured latent spaces for subgoal generation via generative models (Nair et al., 2018; Chane-Sane et al., 2021), contrastive objectives (Zheng et al., 2024; Myers et al., 2024), or expert demonstrations (Konidaris et al., 2012). Similarly to ALPS and PcLast, HILP (Park et al., 2024b) learns a geometric state abstraction preserving the temporal structure of the MDP. SAW (Zhou & Kao, 2026) instead learns a flat policy via advantage-weighted importance sampling bootstrapped on in-trajectory waypoint states. We compare against both HILP and SAW in Appendix B.

### 8. Conclusion

In this paper, we have demonstrated the usage of the Laplacian representation for decision-time planning through the development and evaluation of ALPS. ALPS is a hierarchical decision-time planning algorithm that leverages the Laplacian representation for both subgoal identification and distance-guided trajectory optimization. ALPS successfully plans in continuous, high-dimensional state spaces using only pre-collected offline datasets, and outperforms commonly used model-free baselines on goal-conditioned tasks from OGBench. Through ablation studies, we show that the hierarchical decomposition is critical to ALPS's success, particularly when the offline dataset has limited coverage of the state-action space.

There are several promising directions for advancing the application of the Laplacian representation in hierarchical planning. In Section 6, we demonstrate that a lower-quality low-level planner can be overcome by increasing the number of subgoals used in the high-level planner. Developing a better low-level planner to use with the Laplacian representation could unlock planning at even longer horizons. Another avenue for future work is to investigate stacking the Laplacian representation on-top of a lower level latent space. Similarly to PcLast, which uses ACRO, an intermediate representation could improve the performance of ALPS. A particularly interesting direction concerns the symmetry assumptions underlying the current approach. Learning an asymmetric Laplacian can address this limitation, and the commute time distance used for trajectory optimization can be replaced by the hitting time (Klein & Randić, 1993), which naturally captures the asymmetry of directed transitions.

## Impact Statement

This paper presents work whose goal is to advance the field of Machine Learning. There are many potential societal consequences of our work, none which we feel must be specifically highlighted here.

## Acknowledgements

We thank Diego Gomez for providing an initial version of the ALLO objective, Siddarth Chandersekar, and Calarina Muslimani, along with members of the Reinforcement Learning and Artificial Intelligence (RLAI) lab, for helpful discussions and feedback. We also thank the authors of PcLast for helpful discussions and resolving our queries. Part of this work has taken place in the Intelligent Robot Learning (IRL) Lab and the Representation and Agent-based Learning (ReAL) Lab at the University of Alberta, which are supported in part by the Natural Sciences and Engineering Research Council of Canada (NSERC); Alberta Machine Intelligence Institute (Amii); Canada CIFAR AI Chair Program, Amii; Mitacs; and Alberta Innovates. It was also enabled in part by computational resources provided by the Digital Research Alliance of Canada.

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

# A. Further Background

In this section, we provide further mathematical details for completeness.

## A.1. CTD

For completeness, we include the transformation of CTD to the scaled Laplacian representation following from (Lovász et al., 1993) and (Klein & Randić, 1993). CTD, $c(i, k)$, is defined as the expected number of steps it takes to travel from node $i$ to $k$ and back to $i$ from a random walk (Lovász et al., 1993).

**Theorem A.1.** *(Klein & Randić, 1993) Let $G = (\mathcal{S}, \mathcal{E})$ be a connected, undirected graph with $N = |\mathcal{S}|$ nodes. Let $\mathbf{u}_j$ be the $j$-th eigenvector of the graph Laplacian, $\mathbf{L} = \mathbf{D} - \mathbf{W}$, with corresponding eigenvalue $\lambda_j$. The commute distance between states $i$ and $k$ is given by:*

$$c(i, k) = vol(G) \sum_{j=1}^{N-1} \left( \frac{\mathbf{u}_j(i)}{\sqrt{\lambda_j}} - \frac{\mathbf{u}_j(k)}{\sqrt{\lambda_j}} \right)^2 \tag{5}$$

*Proof.* This result was first established by Klein & Randić (1993) using electrical network theory. The CTD on a graph can be computed using the generalized inverse $\mathbf{L}^\dagger$ of the graph Laplacian $\mathbf{L}$. Since, $\mathbf{L}$ is eigendecomposable, $\mathbf{L}^\dagger = \sum_{j=1}^{n-1} \frac{1}{\lambda_j} \mathbf{u}_j \mathbf{u}_j^\top$. Specifically for computing the commute time, we have:

$$c(i, k) = \text{vol}(G) \left( \mathbf{L}^\dagger(i, i) - 2\mathbf{L}^\dagger(i, k) + \mathbf{L}^\dagger(k, k) \right), \tag{6}$$

where $\text{vol}(G) = \sum_{i=0}^{N-1} \mathbf{d}(i)$ is the volume of the graph. Substituting $\mathbf{L}^\dagger$ into Eq. 6:

$$\begin{aligned}
\mathbf{L}^\dagger(i, i) - 2\mathbf{L}^\dagger(i, k) + \mathbf{L}^\dagger(k, k) &= \sum_{j=1}^{N-1} \frac{1}{\lambda_j} \mathbf{u}_j(i)^2 - 2 \sum_{j=1}^{N-1} \frac{1}{\lambda_j} \mathbf{u}_j(i)\mathbf{u}_j(k) + \sum_{j=1}^{N-1} \frac{1}{\lambda_j} \mathbf{u}_j(k)^2 \\
&= \sum_{j=1}^{N-1} \frac{1}{\lambda_j} \left( \mathbf{u}_j(i)^2 - 2\mathbf{u}_j(i)\mathbf{u}_j(k) + \mathbf{u}_j(k)^2 \right) \\
&= \sum_{j=1}^{N-1} \left( \frac{\mathbf{u}_j(i)}{\sqrt{\lambda_j}} - \frac{\mathbf{u}_j(k)}{\sqrt{\lambda_j}} \right)^2.
\end{aligned}$$

$\square$

To leverage this theoretical result, we define the scaled Laplacian representation vector for a given node $i$, denoted as $\psi(i)$.[6] The $j$-th entry of this representation vector is scaled by the inverse square root of its corresponding eigenvalue: $\psi_j(i) = \mathbf{u}_j(i)/\sqrt{\lambda_j}$. By substituting this definition into our proven result, we see that the commute time distance is exactly proportional to the squared Euclidean distance in this latent space. Since the volume of the graph, $\text{vol}(G)$, is a constant scalar for any static environment, we can write:

$$c(i, k) \propto \|\psi(i) - \psi(k)\|_2^2 = \sum_{j=1}^{N-1} \left( \frac{\mathbf{u}_j(i)}{\sqrt{\lambda_j}} - \frac{\mathbf{u}_j(k)}{\sqrt{\lambda_j}} \right)^2. \tag{7}$$

Hence, the $\psi$-space is isometric to the commute time distance, making it an effective space for trajectory optimization. However, on large graphs, commute time degenerates toward $\left( \frac{1}{\mathbf{d}(i)} + \frac{1}{\mathbf{d}(k)} \right)$, which is dominated by high-eigenvalue terms (Von Luxburg et al., 2014). Truncating to $D$ eigenvectors discards these terms, preserving the global manifold structure. The resulting approximation error is bounded by $O(1/\lambda_{D+1})$, where $\lambda_{D+1}$ is the $(D + 1)$-th eigenvalue of the Laplacian. Also, note that we learn the eigenvectors of the Laplacian corresponding to the discounted transition operator $P_\pi^{\gamma_s} = (1 - \gamma_s) \sum_k (\gamma_s P_\pi)^k$, not the one-step transition operator. As noted in Appendix B of Wu et al. (2019), the generalized multi-step (discounted) formulation provides better performance for RL applications.

---

[6]We refer to this as the scaled Laplacian representation, though it appears under several names in the literature, including the reachability-aware Laplacian representation (Wang et al., 2023a), and the commute-time embedding (Qiu & Hancock, 2007).

# B. Baselines

We briefly explain all the baselines considered in this section.

## B.1. PcLast

Plannable Continuous Latent States  (PcLast; Koul et al., 2024) maps observations to a latent representation and associates neighboring states together by optimizing a contrastive loss:

$$
\arg \min_{\Gamma,\alpha,\beta} - \Bigg( \log \big( \sigma(e^{\alpha} - e^{\beta} \|\Gamma(\xi(s_t)) - \Gamma(\xi(s_{t+d}))\|^2) \big)
$$
$$
+ \log \big( 1 - \sigma(e^{\alpha} - e^{\beta} \|\Gamma(\xi(s_t)) - \Gamma(\xi(s_r))\|^2) \big) \Bigg) \tag{8}
$$

where $\xi$ is a space representation to remove exogenous noise (Islam et al., 2023), $\Gamma$ is the PcLast space, $s_t$ and $s_{t+d}$ are positive pairs $d$ transitions apart, with $s_r$ as a negative example, sampled uniformly from the data buffer, $\alpha$ and $\beta$ are the hyperparameters.

## B.2. OGBench Baselines

Goal-Conditioned Behavior Cloning (Lynch et al., 2020; Ghosh et al., 2021) is a baseline that performs behavior cloning by sampling a future state from the same trajectory as the goal. Goal-Conditioned Implicit {V, Q}-Learning are goal-conditioned variants of implicit Q-Learning  (IQL; Kostrikov et al., 2022) that fits optimal value functions (V* and Q*) using expectile regression. Quasimetric RL (Wang et al., 2023b) learns a quasimetric distance function that satisfies the triangle inequality and uses this to represent goal-conditioned value functions. Contrastive RL (Eysenbach et al., 2022) uses contrastive learning to learn a goal-conditioned value function. Hierarchical Implicit Q-Learning (Park et al., 2023) is a hierarchical model-free algorithm where the high-level predicts the representation of an optimal $k$-step subgoal, and the low-level policy predicts the optimal action for this subgoal.

## B.3. Other Baselines

HILP (Park et al., 2024b) learns a geometric state abstraction that preserves the temporal structure of the MDP for zero-shot goal-conditioned planning. SAW (Zhou & Kao, 2026) learns a flat policy via advantage-weighted importance sampling over in-trajectory waypoint states. RLDP (Jajoo et al., 2026) learns a latent-predictive state encoder used as a task encoder for successor features trained via a contrastive loss. TD-JEPA (Bagatella et al., 2026) uses TD learning to train latent-predictive representations of long-term policy dynamics, learning policies directly in the resulting latent space. Since these methods were evaluated on OGBench, we report their results directly from the respective papers.

*Table 4.* **Success rate (%) on a subset of locomotion and manipulation tasks from OGBench.** The results are averaged over 8 seeds (4 seeds for pixel-based tasks), and we report the standard deviation after the ± sign.

| Dataset | HILP | SAW | RLDP | TD-JEPA | ALPS |
|---|---|---|---|---|---|
| antmaze-medium-navigate-v0 | $84 \pm 3$ | $97 \pm 1$ | $75 \pm 4$ | $70 \pm 4$ | $97 \pm 2$ |
| antmaze-medium-stitch-v0 | $51 \pm 2$ | $96 \pm 1$ | $58 \pm 3$ | $62 \pm 5$ | $93 \pm 7$ |
| antmaze-large-navigate-v0 | $53 \pm 4$ | $90 \pm 3$ | $36 \pm 5$ | $57 \pm 4$ | $93 \pm 5$ |
| antmaze-large-stitch-v0 | $12 \pm 2$ | $66 \pm 9$ | $20 \pm 3$ | $41 \pm 3$ | $95 \pm 2$ |
| antmaze-medium-explore-v0 | $2 \pm 1$ | $25 \pm 4$ | $5 \pm 2$ | $20 \pm 2$ | $100 \pm 0$ |
| cube-single-play-v0 | $74 \pm 4$ | $72 \pm 5$ | $20 \pm 2$ | $34 \pm 3$ | $68 \pm 6$ |
| scene-play-v0 | $44 \pm 2$ | $63 \pm 6$ | $12 \pm 2$ | $38 \pm 1$ | $26 \pm 3$ |
| visual-antmaze-medium-navigate-v0 | $85 \pm 4$ | $95 \pm 0$ | $98 \pm 1$ | $97 \pm 1$ | $94 \pm 5$ |
| visual-antmaze-large-navigate-v0 | $47 \pm 4$ | $82 \pm 4$ | $64 \pm 4$ | $75 \pm 3$ | $88 \pm 3$ |

# C. Statistical testing

Results reported in Section 5 on OGBench are statistically tested using the Wilcoxon signed-rank test. For each baseline, ALPS and the baseline are compared across datasets to show the two algorithms aren't drawing from the same distribution. Because we are testing across multiple baselines with a single sample of ALPS we use the Holm-Bonferroni corrections.

We set an acceptable error rate of $p < 0.001$, which our tests show was too cautious in Table 5.

*Table 5.* Wilcoxon signed-rank test $p$-values (two-sided) with Holm-Bonferroni correction comparing ALPS against baselines.

|  | GCBC | GCIVL | GCIQL | QRL | CRL | HIQL |
|---|---|---|---|---|---|---|
| $p$-value | 0.000001 | 0.000002 | 0.000001 | 0.000001 | 0.000002 | 0.000031 |

## D. Cross-Entropy Method for Decision-time Planning

The Cross-Entropy Method (CEM; Rubinstein, 1997) is a popular MPC-based planner that has been successful in model-based RL settings (Finn & Levine, 2017; Chua et al., 2018; Hafner et al., 2019; Pinneri et al., 2021; Gürtler & Martius, 2025). CEM begins by sampling action sequences from a normal distribution and updating its mean and variance based on the lowest-cost trajectories. We used the behavior prior, $\pi_{\text{prior}}$, to generate the initial action sequence. $\pi_{\text{prior}}$ predicts an action $\hat{A}_t$ in a state $S_t$, which is then fed to the forward model $f$ to predict the next state $\hat{S}_{t+1}$. This predicted state $\hat{S}_{t+1}$ is then fed back to $\pi_{\text{prior}}$ to predict $\hat{A}_{t+1}$. This process repeats until the planning horizon $H$, resulting in an action sequence $\mathbf{a}_{t:t+H-1}$. Correlated noise is then added to the initial action sequence to generate multiple action sequences. CEM uses these action sequences to generate multiple trajectories, whose costs are computed in scaled Laplacian representation, $\psi$-space, because it is isometric to CTD. The CEM algorithm is outlined in Algorithm 2.

---

**Algorithm 2** Cross-Entropy Method Planner

---

**Input**: $s$, $z_{\text{sub}}$, $\psi$, $f$, $\pi_{\text{prior}}$
**CEM parameters:** planner horizon $H$, iterations $N_{\text{iter}}$, number of samples $N_{\text{s}}$, elite ratio $N_{\text{e}}$
=== Warm Start CEM with Policy ===
$\mathbf{a}_{1:H} \leftarrow$ rollout $\pi_{\text{prior}}$ auto-regressively with $f$ from $s$
=== CEM iterations ===
**for** $i = 1$ to $N_{\text{iter}}$ **do**
    sample $\{\mathbf{a}_{1:H}^j\}_{j=1}^{N_{\text{s}}}$ around $\mathbf{a}_{1:H}$          ▷ sample action sequences
    === Cost Computation ===
    **for** $j = 1$ to $N_{\text{s}}$ **do**
        $\{\hat{S}_t\}_{t=1}^H \leftarrow f(s, \mathbf{a}_{1:H}^j)$          ▷ rollout action sequences using $f$
        $J^m \leftarrow \sum_{t=1}^H (\|\psi(\hat{S}_t^m) - z_{\text{sub}}\| + \lambda \|\mathbf{a}_t^m\|^2)$
    **end for**
    $\mathbf{a}_{1:H} \leftarrow$ mean of top-$N_{\text{e}}$ elite trajectories          ▷ update sampling distribution
**end for**
**return** $\mathbf{a}_{1:H}$

---

## E. Environments

In this section, we will discuss all the environments that were considered for evaluation of ALPS.

### E.1. Maze2D—PointMass Environments

We introduce the *Maze2D—PointMass* environments here, introduced in (Koul et al., 2024). Each maze is a unit square with varied wall configurations. The agent controls a point mass with actions corresponding to the coordinate space change $(\Delta x, \Delta y)$ bounded by the range $[-0.2, 0.2]$ for each action. Observations are defined as a single-channel $(100 \times 100)$ image encoding the current position of the agent and no other environmental information. A Gaussian blur ($\sigma = 1.0$) is applied to the agent's coordinate position, and the resulting image is normalized to $[0, 1]$ (Fig. 4a). In the presence of obstacles, the point mass starts from $s_t$ and is moved along the direction of $a_t$ until it collides with an obstacle. We consider all three variants: *Hallway*, *Rooms*, and *Spiral* of the Maze environment, whose layouts are shown in Fig. 4. An offline dataset of $500K$ transitions is generated using a uniform random policy. We follow PcLast's empirical design where the agent must navigate from a starting position to within $0.03$ units of a known target position within 30 actions.

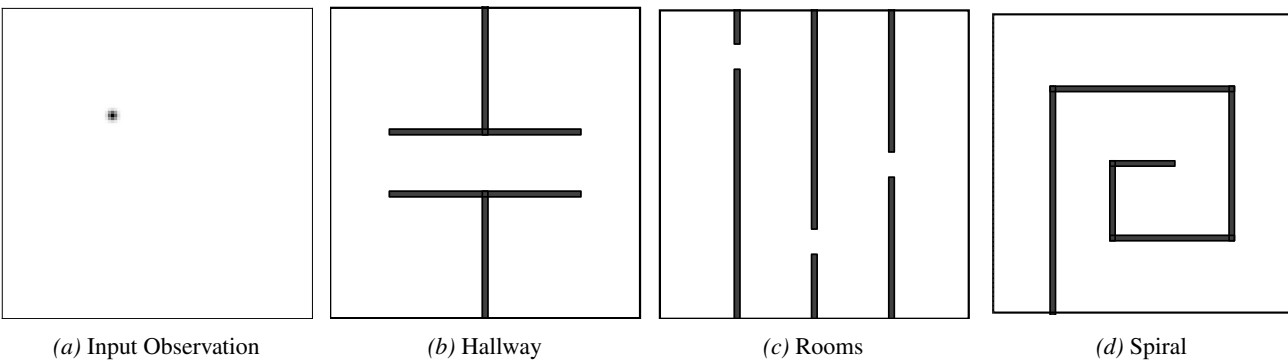

*(a)* Input Observation    *(b)* Hallway    *(c)* Rooms    *(d)* Spiral

*Figure 4.* (a) Input observation, (b)-(d) 2-D Maze environments.

### E.2. OGBench Environments

We use three locomotion tasks from OGBench: *pointmaze*, *antmaze*, and *humanoidmaze*. They require controlling 2-DoF *ball*, 8-DoF *ant*, and 21-DoF *humanoid* bodies, respectively. We consider both state-based and pixel-based variants of these locomotion-based tasks. In state-based variants, the agent has access to the full low-dimensional state representation, including its current $x$-$y$ position. In pixel-based variants, the agent only receives $64 \times 64 \times 3$ images rendered from a third-person camera viewpoint. To avoid the need for recurrent networks, the floor is colored to enable the agent to infer its location directly from the images.

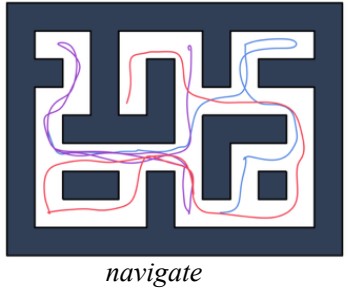 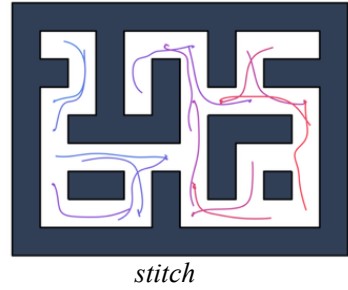 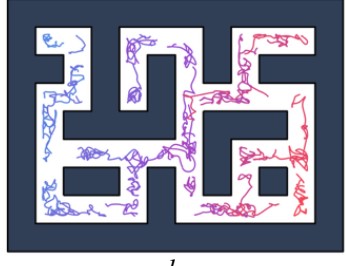

*navigate*          *stitch*          *explore*

*Figure 5.* OGBench dataset variants.

There are three dataset variants, each collected through a directional policy trained via SAC (Haarnoja et al., 2018) and a high-level waypoint controller. The dataset types are as follows: (i) *navigate*, collected by a noisy expert policy that navigates the maze by repeatedly reaching randomly sampled goals, (ii) *stitch*, collected through shorter goal-reaching trajectories from a noisy expert, which test the agent's stitching ability, and (iii) *explore*, collected by commanding the low-level policy with a large amount of action noise, aimed to test agent's navigation skills from extremely low-quality (yet high-coverage) data. When collecting datasets, Gaussian noise with a standard deviation of 0.5 (*pointmaze*), 1.0 (*explore*), or 0.2 (others) is added to the expert actions. Fig. 5 shows example trajectories in these datasets.

All the mazes considered in OGBench for locomotion tasks are depicted in Fig. 6. It consists of four mazes: *medium*, *large*, *giant*, and *teleport*. *medium* and *large* maze have the same layouts as in D4RL (Fu et al., 2020). *giant* maze is the largest maze, twice the size of *large*, and is primarily designed to test the agent's long-horizon reasoning, i.e., capability of navigating from a starting state to a goal state that is many steps apart. *teleport* is a stochastic environment with teleportation gates: black hole as *teleport-in* and white hole as *teleport-out*. If the agent enters the black hole, it is randomly spawned at any of the white holes. One of the white holes is a dead end; thus, accessing the teleport holes is risky. Hence, the agent must learn to avoid black holes to complete the task reliably.

We also consider two variants of robotic manipulation tasks: *Cube* and *Scene*. These tasks are designed to test the agent's object manipulation, sequential generalization, and combinatorial generalization abilities. These environments come with a 6-DoF UR5e robot arm and Robotiq $2F-85$ gripper model from MuJoCo, controlled with a 5-D action space end-effector. The dataset is collected by non-Markovian scripted policies with temporally correlated noise (Park et al., 2025).

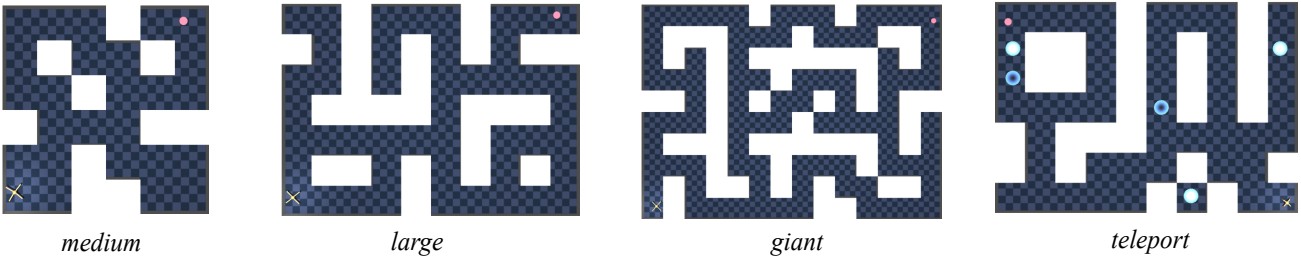

| medium | large | giant | teleport |

*Figure 6.* OGBench mazes.

*Cube* tasks involve the pick-and-place manipulation of cube blocks, with the goal of controlling the robotic arm to arrange the cubes into a desired configuration. We consider two variants: *single* and *double*, in which the agent is required to move, swap, stack, and permute the cube blocks. *Scene* tasks involve manipulating the robotic arm to press the button to toggle lock states, pick up the cube, and put it in the drawer. This task is designed to challenge the sequential, long-horizon reasoning capabilities, ranging from a single atomic (e.g., a single pick-and-place) to the longest task involving eight atomic behaviors. Hence, the agent must be able to plan and sequentially combine the learned manipulation skills. The layout for *cube* and *scene* is shown in Fig. 7.

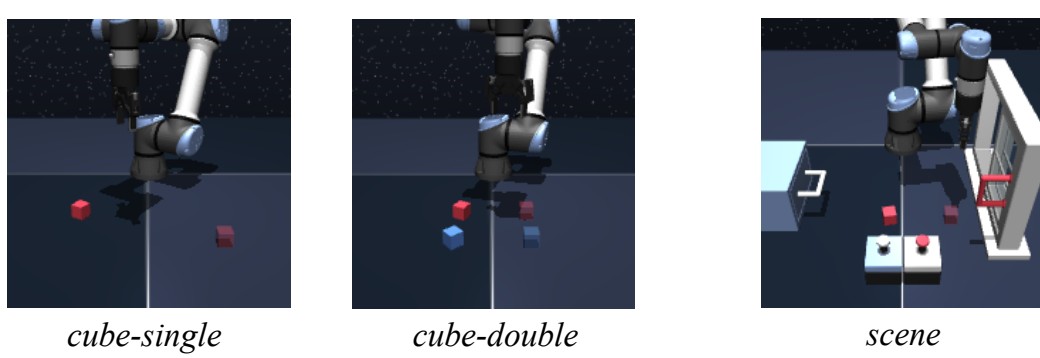

| cube-single | cube-double | scene |

*Figure 7.* OGBench manipulation environments.

*Table 6.* Environment specifications.

| Environment Type | Environment | State Dim | Action Dim | Max Episode Length |
|---|---|---|---|---|
| pointmaze | pointmaze-medium-v0 | 2 | 2 | 1000 |
| | pointmaze-large-v0 | 2 | 2 | 1000 |
| | pointmaze-giant-v0 | 2 | 2 | 1000 |
| | pointmaze-teleport-v0 | 2 | 2 | 1000 |
| antmaze | antmaze-medium-v0 | 29 | 8 | 1000 |
| | antmaze-large-v0 | 29 | 8 | 1000 |
| | antmaze-giant-v0 | 29 | 8 | 1000 |
| | antmaze-teleport-v0 | 29 | 8 | 1000 |
| humanoidmaze | humanoidmaze-medium-v0 | 69 | 21 | 2000 |
| | humanoidmaze-large-v0 | 69 | 21 | 2000 |
| | humanoidmaze-giant-v0 | 69 | 21 | 4000 |
| visual-antmaze | visual-antmaze-medium-v0 | $64 \times 64 \times 3$ | 8 | 1000 |
| | visual-antmaze-large-v0 | $64 \times 64 \times 3$ | 8 | 1000 |
| | visual-antmaze-giant-v0 | $64 \times 64 \times 3$ | 8 | 1000 |
| | visual-antmaze-teleport-v0 | $64 \times 64 \times 3$ | 8 | 1000 |
| cube | cube-single-v0 | 28 | 5 | 200 |
| | cube-double-v0 | 37 | 5 | 500 |
| scene | scene-v0 | 40 | 5 | 750 |

An episode ends as soon as the agent reaches the proximity of the goal location that defines success, or when the episode ends, which varies across environments. For locomotion tasks, joint positions are not used to determine success, unlike previous works (Park et al., 2023). As mentioned in the OGBench, for manipulation tasks, success criteria are based solely on the object configurations; the arm pose is not considered in determining success. Specifically, this proximity value is 0.5

units for *pointmaze*, 1 unit for *antmaze/humanoidmaze*, 0.04 units for *cube* and *scene* individual tasks. Since the number of environmental steps and the dataset collection procedure differ across environments, we report both the environment and dataset specifications in Tables 6 and 7, respectively.

Each task in OGBench has 5 pre-defined start-goal pairs, which are slightly randomized on every reset. Following the OGBench protocols, the performance of each agent is averaged over 750 evaluations (3 evaluation epochs $\times$ 5 test-time goals $\times$ 50 rollouts). Since ALPS involves sequential training, we adopt the following evaluation protocol. Since ALPS is not an end-to-end algorithm, for a fair comparison with the rest of the algorithms, the Laplacian representation from ALLO is first pre-trained and frozen at three distinct checkpoints (800K, 900K, and 1M steps for state-based tasks and 300K, 400K, and 500K steps for pixel-based tasks). Then, behavior prior is trained with these weights for the corresponding time steps. All the OGBench baselines are directly evaluated at 800K, 900K, and 1M steps for state-based tasks and 300K, 400K, and 500K steps for pixel-based tasks. The final performance is averaged across these three evaluation stages and 8 random seeds for state-based tasks and 4 random seeds for pixel-based tasks.

*Table 7.* Dataset specifications.

| Environment Type | Dataset Type | Dataset | # Transitions | # Episodes | Data Episode Length |
|---|---|---|---|---|---|
| pointmaze | navigate | pointmaze-medium-navigate-v0 | 1M | 1000 | 1000 |
| | | pointmaze-large-navigate-v0 | 1M | 1000 | 1000 |
| | | pointmaze-giant-navigate-v0 | 1M | 500 | 2000 |
| | | pointmaze-teleport-navigate-v0 | 1M | 1000 | 1000 |
| | stitch | pointmaze-medium-stitch-v0 | 1M | 5000 | 200 |
| | | pointmaze-large-stitch-v0 | 1M | 5000 | 200 |
| | | pointmaze-giant-stitch-v0 | 1M | 5000 | 200 |
| | | pointmaze-teleport-stitch-v0 | 1M | 5000 | 200 |
| antmaze | navigate | antmaze-medium-navigate-v0 | 1M | 1000 | 1000 |
| | | antmaze-large-navigate-v0 | 1M | 1000 | 1000 |
| | | antmaze-giant-navigate-v0 | 1M | 500 | 2000 |
| | | antmaze-teleport-navigate-v0 | 1M | 1000 | 1000 |
| | stitch | antmaze-medium-stitch-v0 | 1M | 5000 | 200 |
| | | antmaze-large-stitch-v0 | 1M | 5000 | 200 |
| | | antmaze-giant-stitch-v0 | 1M | 5000 | 200 |
| | | antmaze-teleport-stitch-v0 | 1M | 5000 | 200 |
| | explore | antmaze-medium-explore-v0 | 5M | 10000 | 500 |
| | | antmaze-large-explore-v0 | 5M | 10000 | 500 |
| | | antmaze-teleport-explore-v0 | 5M | 10000 | 500 |
| humanoidmaze | navigate | humanoidmaze-medium-navigate-v0 | 2M | 1000 | 2000 |
| | | humanoidmaze-large-navigate-v0 | 2M | 1000 | 2000 |
| | | humanoidmaze-giant-navigate-v0 | 4M | 1000 | 4000 |
| | stitch | humanoidmaze-medium-stitch-v0 | 2M | 5000 | 400 |
| | | humanoidmaze-large-stitch-v0 | 2M | 5000 | 400 |
| | | humanoidmaze-giant-stitch-v0 | 4M | 10000 | 400 |
| visual-antmaze | navigate | visual-antmaze-medium-navigate-v0 | 1M | 1000 | 1000 |
| | | visual-antmaze-large-navigate-v0 | 1M | 1000 | 1000 |
| | | visual-antmaze-giant-navigate-v0 | 1M | 500 | 2000 |
| | | visual-antmaze-teleport-navigate-v0 | 1M | 1000 | 1000 |
| | stitch | visual-antmaze-medium-stitch-v0 | 1M | 5000 | 200 |
| | | visual-antmaze-large-stitch-v0 | 1M | 5000 | 200 |
| | | visual-antmaze-giant-stitch-v0 | 1M | 5000 | 200 |
| | | visual-antmaze-teleport-stitch-v0 | 1M | 5000 | 200 |
| | explore | visual-antmaze-medium-explore-v0 | 5M | 10000 | 500 |
| | | visual-antmaze-large-explore-v0 | 5M | 10000 | 500 |
| | | visual-antmaze-teleport-explore-v0 | 5M | 10000 | 500 |
| cube | play | cube-single-play-v0 | 1M | 1000 | 1000 |
| | | cube-double-play-v0 | 1M | 1000 | 1000 |
| scene | play | scene-play-v0 | 1M | 1000 | 1000 |

# F. Implementation details

We use the same architecture for all domains. The ALLO encoder $\phi$ is a four-layer MLP network with hidden layers of size 256. The first layer has Layer norm (Ba et al., 2016) followed by a tanh activation, while all other layers have ReLU activations. The network outputs 32 eigenvectors, decided by a sweep over 16, 32, and 64. The forward model $f$ network is a four-layer MLP with hidden layers of size 512 and ReLU activations. The behavior prior $\pi_{\text{prior}}$ is a four-layer MLP with

hidden size 512 and ReLU activations. For image states, three convolutional layers are prepended to all networks.

For training the ALLO encoder, we set the geometric distribution parameter $\gamma_s = 0.2$ for PcLast environments, $\gamma_s = 0.6$ for OGBench locomotions tasks, and $\gamma_s = 0.2$ for OGBench manipulation tasks. All other hyperparameters are set to the default values from Gomez et al. (2023) and listed in Table 8. These values are chosen through a sweep over values in the range $[0.2, 0.8]$ with a stride of 0.1. Unless otherwise specified, the horizon used for the behavior prior is $k \sim U[1, 50]$, and the max horizon for the forward model $H_{\mathrm{f}} = 10$. All networks are trained using Adam optimizer (Kingma & Ba, 2015) with a step size of $\alpha = 10^{-4}$ for $\phi$ chosen as the default from (Gomez et al., 2023) and $\alpha = 3 \times 10^{-4}$ for $f$ and $\pi_{\mathrm{prior}}$ with a batch size of 1024 (256 for pixel-based tasks). We only tuned the depth of the neural networks. For domains with continuous states, we centered the values to have a zero mean and a unit standard deviation. For image domains, we normalize the values to be in the range $[0, 1]$. All the neural networks are trained for 1M steps for state-based tasks and 500K for image-based tasks. The number of training steps and the batch size are chosen the same as in the OGBench implementation to ensure a fair comparison.

*Table 8.* Hyperparameters of ALPS.

| Hyperparameter | Value |
|---|---|
| === *ALLO Parameters* === | |
| Sampling discount ($\gamma_s$) | 0.6 (ogbench locomotion), 0.2 (ogbench manipulation, *Maze*) |
| Number of eigenvectors | 32 |
| ALLO learning rate | $1 \times 10^{-4}$ |
| Duals initial value | $-1.0$ |
| Barrier initial value | 0.5 |
| Min duals | $-100.0$ |
| Max duals | 100.0 |
| Min barrier coefficients | 0.0 |
| Max barrier coefficients | 0.5 |
| Step size duals | 1.0 |
| ALLO training steps | $1 \times 10^6$ (states), $5 \times 10^5$ (pixels) |
| === *Forward Model Parameters* === | |
| Multistep Dynamics Horizon ($H_{\mathrm{f}}$) | 10(ogbench), 3 (*Maze*) |
| Dynamics learning rate | $3 \times 10^{-4}$ |
| Dynamics training steps | $1 \times 10^6$ (states), $5 \times 10^5$ (pixels) |
| === *Behavior Prior Parameters* === | |
| Prior max horizon ($K_{\mathrm{max}}$) | 50 |
| Prior learning rate | $3 \times 10^{-4}$ |
| Prior training steps | $1 \times 10^6$ (states), $5 \times 10^5$ (pixels) |
| === *Planner Parameters* === | |
| Number of clusters | 64 (*medium*), 96 (*large, teleport*), 128 (*giant*) |
| Top-$p$ | 0.95 |
| Noise beta ($\beta$) | 0.9 |
| Sigma ($\sigma$) | 0.5 |
| Penalty Coefficient ($\lambda$) | 0.01 |
| Planner Momentum ($\eta$) | 0.3 |
| Planner Horizon ($H$) | 20 (ogbench), 2(*Maze*) |
| CEM Iterations ($N_{\mathrm{iter}}$) | 5 |
| CEM Samples ($N_{\mathrm{s}}$) | 500 |
| CEM Elite ratio ($N_{\mathrm{e}}$) | 0.15 |

The number of clusters in the cluster graph is dependent on the maze size for the locomotion tasks: 64 (*medium*), 96 (*large, teleport*), and 128 (*giant*). For the manipulation tasks, the number of clusters is set to 8. To construct the connections of the cluster graph, nucleus sampling with a $p$-value of 0.95 is used, inspired by Koul et al. (2024).

All planner parameters are consistent across all environments. The planning horizon $H$ for the CEM is set to be 20, swept over $\{10, 20\}$ on all tasks. We add temporally-correlated noise with $\beta = 0.9$ and magnitude $\sigma = 0.5$ to create $N_{\mathrm{s}} = 500$ trajectories around the mean trajectory rolled out from $\pi_{\mathrm{prior}}$. The actions are refined for $N_{\mathrm{iter}} = 5$ iterations, updating the distribution using the top $N_{\mathrm{e}} = 15\%$ samples as elites. A momentum $\eta$ of 0.3 is used to smooth out updates to the final mean trajectory. These values were selected by a grid search over $N_{\mathrm{s}} \in \{200, 500\}, \sigma \in \{0.3, 0.5, 0.7\}, \eta \in \{0.3, 0.5, 0.7\}$. The penalty coefficient $\lambda$ for penalizing large actions in the cost function is set to 0.01, selected from $\{0.01, 0.05, 0.1\}$.

# G. Further Results and Visualizations

Here, we provide some additional results and visualizations to better understand ALPS.

## G.1. Visualization for PcLast Environments

We further show the effectiveness of the Laplacian representation using the Maze2D—PointMass Spiral environment (Fig. 4d). As shown in Fig. 8, spectral clustering correctly partitions the spiral environment into well-connected regions respecting the wall boundaries. Crucially, the learned representation captures CTD effectively. For instance, the inner part of the spiral is geometrically closer to the reference state but far in terms of temporal distance. Further, the topological accuracy provides a smooth gradient for trajectory optimization, validating ALPS's strong performance in these Maze2D environments with only a low-level planner, as reported in Table 1.

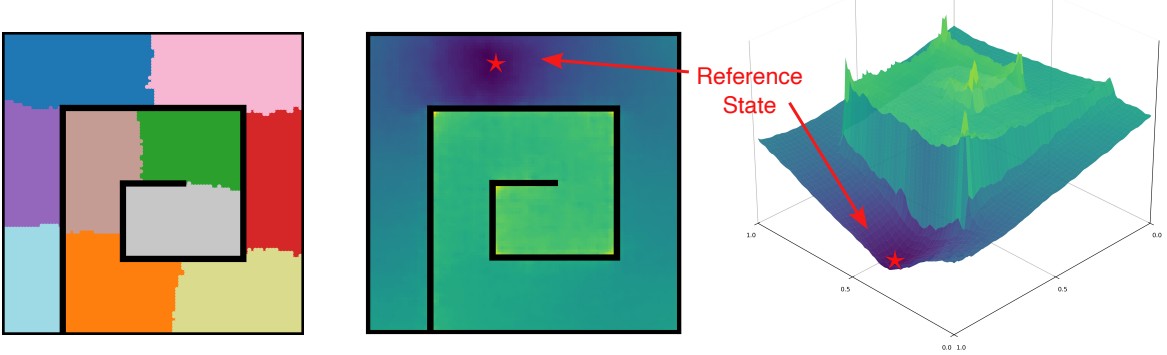

*Figure 8.* Visualization of the $\psi$-space properties in the spiral maze environment (Fig. 4d). (**left**) Cluster labels assigned to each state in the dataset via clustering in $\psi$-space.(**center**) Heatmap of $c(s^\star, s_i)$ distance from a reference state ($s^\star$ denoted by $\star$ in the figure) to each state in the dataset. (**right**) 3-D visualization of the distance landscape from the same reference state. This further validates the CTD property and spectral clustering characteristic of $\psi$-space.

## G.2. Comparison with PcLast

In Section 5, we compared ALPS[†] and PcLast to the best of our ability using the code provided by the authors. When initially testing Plannable Continuous Latent States (PcLast) on the Maze2D-Point Mass environments without a gaussian blur, we were unable to reproduce their results. In Table 9, we report the results we were able to reproduce using their code and discussion with the authors and the original performance reported in Koul et al. (2024).

*Table 9.* Further results using PcLast on Maze2D-Point Mass domain. We follow the same protocol as discussed in Section 5.

| Environment | Clusters | PcLast (Reported) | PcLast (Ours) |
|:---:|:---:|:---:|:---:|
| Hallway | 1 | $88 \pm 3$ | $40 \pm 2$ |
| | 16 | $97 \pm 5$ | $69 \pm 6$ |
| Rooms | 1 | $69 \pm 3$ | $26 \pm 3$ |
| | 16 | $90 \pm 10$ | $36 \pm 7$ |
| Spiral | 1 | $50 \pm 4$ | $39 \pm 4$ |
| | 16 | $89 \pm 10$ | $60 \pm 7$ |

## G.3. Ablation on number of eigenvectors

For a varying number of eigenvectors, we tested ALPS across $D$ (2, 8, 16, 24, 32, 48) in antmaze-large/giant environments, while keeping the default number of clusters fixed. The results are shown in Table 10. Performance largely saturates around $D = 24 - 32$, indicating that the coarse spatial structure of the environment is well captured by a modest number of eigenvectors. Note that $D = 2$ fails almost entirely, confirming that the multiple time scale information encoded by additional eigenvectors is essential for the success of ALPS.

*Table 10.* Ablation on number of eigenvectors.

| Dataset / #eigenvectors | 2 | 8 | 16 | 24 | 32 | 48 |
|---|---|---|---|---|---|---|
| antmaze-large-navigate-v0 | $33 \pm 6$ | $90 \pm 2$ | $94 \pm 1$ | $93 \pm 8$ | $94 \pm 3$ | $94 \pm 2$ |
| antmaze-large-stitch-v0 | $19 \pm 6$ | $93 \pm 5$ | $95 \pm 3$ | $94 \pm 5$ | $95 \pm 2$ | $94 \pm 4$ |
| antmaze-giant-navigate-v0 | $0 \pm 0$ | $28 \pm 7$ | $51 \pm 14$ | $64 \pm 12$ | $68 \pm 5$ | $76 \pm 4$ |
| antmaze-giant-stitch-v0 | $1 \pm 1$ | $59 \pm 10$ | $88 \pm 3$ | $91 \pm 2$ | $92 \pm 2$ | $92 \pm 2$ |

### G.4. Teleport Environment

As discussed in the Section 5.3, *teleport* mazes pose an interesting problem in the context of the Laplacian representation. Since ALLO objective learns a symmetrized version, in $\psi$-space, the embeddings of teleport in and teleport out gates are close to each other. We verify this empirically in Fig. 9 where all the teleport gates are getting clustered to the same cluster. Thus, if a goal is closer to the teleport gates, the agent attempts to access them, which is suboptimal due to the risk of entering an inescapable region.

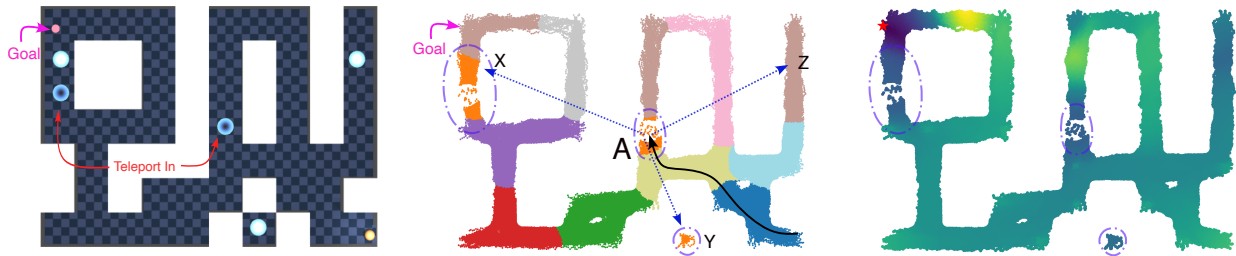

*Figure 9.* Visualization of the cluster assignment and Laplacian space distance for the *pointmaze-teleport* maze environment from OGBench. In this task, the goal lies closer to teleport-out gate. Since all the teleport gates are clustered together as their embeddings lies together, *ball* tries to access the teleport gate after reaching point A. However, *ball* may end up at point Y through which it cannot escape.

### G.5. Individual Task Results

We report individual evaluation goals success rates for each task averaged over 150 rollouts (3 evaluation epochs $\times 50$ rollouts) for *pointmaze* (Table 11), *antmaze* (Table 12), *humanoidmaze* (Table 13), *cube* (Table 14), *scene* (Table 15), and *visual-antmaze* (Table 16).

### G.6. Ablations

For planning without the $\pi_{\text{prior}}$ case, CEM must find optimal actions from scratch. We increase the temporally-correlated noise magnitude $\sigma = 1.0$ because CEM has to search across the entire action space. Number of CEM iterations $N_{\text{iter}}$ is set to 15, selected from a sweep over $\{5, 10, 15, 20\}$, as CEM takes more time to converge now. The number of CEM samples $N_{\text{s}}$ is also increased to 5000 as the search space is wider compared to the case when it can access the $\pi_{\text{prior}}$ case, selected with a grid search over $\{500, 1000, 2000, 5000, 10000\}$. This further shows that having access to $\pi_{\text{prior}}$ can accelerate the convergence of CEM with a smaller number of samples.

We report the success rates (%) of all six variants of ALPS, discussed in Section 6, in Table 17.

*Table 11.* Full Results on *pointmaze*.

| Environment Type | Dataset Type | Dataset | Task | GCBC | GCIVL | GCIQL | QRL | CRL | HIQL | ALPS |
|---|---|---|---|---|---|---|---|---|---|---|
| pointmaze | navigate | pointmaze-medium-navigate-v0 | task1 | 30 ± 27 | 88 ± 16 | 97 ± 4 | 100 ± 0 | 20 ± 6 | 99 ± 1 | 85 ± 8 |
| | | | task2 | 3 ± 2 | 95 ± 10 | 76 ± 29 | 94 ± 17 | 45 ± 25 | 87 ± 7 | 96 ± 3 |
| | | | task3 | 5 ± 5 | 37 ± 28 | 10 ± 28 | 23 ± 20 | 30 ± 4 | 55 ± 13 | 84 ± 9 |
| | | | task4 | 0 ± 1 | 2 ± 2 | 0 ± 0 | 94 ± 14 | 28 ± 29 | 82 ± 12 | 88 ± 7 |
| | | | task5 | 4 ± 3 | 92 ± 7 | 79 ± 6 | 97 ± 8 | 24 ± 13 | 70 ± 10 | 83 ± 26 |
| | | | overall | 9 ± 6 | 63 ± 6 | 53 ± 8 | 82 ± 5 | 29 ± 7 | 79 ± 5 | 82 ± 10 |
| | | pointmaze-large-navigate-v0 | task1 | 63 ± 11 | 76 ± 23 | 86 ± 14 | 95 ± 8 | 42 ± 27 | 83 ± 13 | 72 ± 23 |
| | | | task2 | 1 ± 2 | 0 ± 0 | 0 ± 0 | 100 ± 0 | 31 ± 24 | 2 ± 7 | 90 ± 8 |
| | | | task3 | 10 ± 7 | 98 ± 5 | 83 ± 8 | 40 ± 50 | 78 ± 7 | 88 ± 10 | 96 ± 3 |
| | | | task4 | 20 ± 18 | 0 ± 0 | 0 ± 0 | 96 ± 7 | 24 ± 14 | 72 ± 19 | 76 ± 23 |
| | | | task5 | 52 ± 17 | 53 ± 20 | 0 ± 0 | 96 ± 7 | 20 ± 10 | 46 ± 16 | 74 ± 21 |
| | | | overall | 29 ± 6 | 45 ± 5 | 34 ± 3 | 86 ± 9 | 39 ± 7 | 58 ± 5 | 80 ± 8 |
| | | pointmaze-giant-navigate-v0 | task1 | 1 ± 3 | 0 ± 0 | 0 ± 0 | 98 ± 7 | 6 ± 15 | 0 ± 0 | 4 ± 5 |
| | | | task2 | 1 ± 4 | 0 ± 0 | 0 ± 0 | 92 ± 16 | 28 ± 10 | 72 ± 17 | 90 ± 22 |
| | | | task3 | 0 ± 0 | 0 ± 1 | 0 ± 0 | 68 ± 27 | 9 ± 5 | 32 ± 11 | 96 ± 4 |
| | | | task4 | 0 ± 0 | 0 ± 0 | 0 ± 0 | 66 ± 20 | 64 ± 17 | 60 ± 22 | 26 ± 19 |
| | | | task5 | 5 ± 12 | 0 ± 0 | 0 ± 0 | 19 ± 32 | 29 ± 28 | 66 ± 20 | 98 ± 2 |
| | | | overall | 1 ± 2 | 0 ± 0 | 0 ± 0 | 68 ± 7 | 27 ± 10 | 46 ± 9 | 67 ± 11 |
| | | pointmaze-teleport-navigate-v0 | task1 | 1 ± 2 | 33 ± 12 | 0 ± 1 | 0 ± 0 | 3 ± 3 | 5 ± 5 | 36 ± 15 |
| | | | task2 | 4 ± 6 | 49 ± 2 | 39 ± 14 | 8 ± 10 | 30 ± 23 | 6 ± 6 | 34 ± 18 |
| | | | task3 | 50 ± 4 | 46 ± 5 | 31 ± 19 | 2 ± 5 | 26 ± 6 | 39 ± 9 | 48 ± 6 |
| | | | task4 | 33 ± 13 | 49 ± 4 | 42 ± 13 | 12 ± 16 | 40 ± 11 | 24 ± 11 | 50 ± 9 |
| | | | task5 | 38 ± 6 | 48 ± 4 | 9 ± 9 | 1 ± 2 | 20 ± 15 | 17 ± 8 | 42 ± 9 |
| | | | overall | 25 ± 3 | 45 ± 3 | 24 ± 7 | 4 ± 4 | 24 ± 6 | 18 ± 4 | 40 ± 6 |
| | stitch | pointmaze-medium-stitch-v0 | task1 | 21 ± 29 | 76 ± 14 | 56 ± 24 | 94 ± 13 | 0 ± 0 | 77 ± 14 | 99 ± 2 |
| | | | task2 | 32 ± 35 | 79 ± 23 | 26 ± 19 | 81 ± 34 | 0 ± 0 | 61 ± 23 | 91 ± 17 |
| | | | task3 | 33 ± 34 | 69 ± 16 | 0 ± 0 | 66 ± 29 | 2 ± 3 | 82 ± 13 | 96 ± 3 |
| | | | task4 | 0 ± 0 | 41 ± 37 | 0 ± 0 | 68 ± 32 | 0 ± 0 | 92 ± 6 | 98 ± 3 |
| | | | task5 | 29 ± 37 | 84 ± 11 | 22 ± 22 | 92 ± 9 | 0 ± 0 | 59 ± 9 | 95 ± 8 |
| | | | overall | 23 ± 18 | 70 ± 14 | 21 ± 9 | 80 ± 12 | 0 ± 1 | 74 ± 6 | 94 ± 6 |
| | | pointmaze-large-stitch-v0 | task1 | 8 ± 13 | 0 ± 1 | 56 ± 11 | 100 ± 1 | 0 ± 0 | 3 ± 5 | 95 ± 3 |
| | | | task2 | 0 ± 0 | 0 ± 0 | 0 ± 0 | 74 ± 37 | 0 ± 0 | 0 ± 0 | 96 ± 3 |
| | | | task3 | 26 ± 28 | 60 ± 29 | 98 ± 4 | 74 ± 23 | 0 ± 0 | 59 ± 25 | 97 ± 2 |
| | | | task4 | 0 ± 0 | 0 ± 0 | 0 ± 0 | 88 ± 32 | 0 ± 0 | 1 ± 4 | 91 ± 6 |
| | | | task5 | 0 ± 0 | 0 ± 0 | 0 ± 0 | 85 ± 22 | 0 ± 0 | 0 ± 0 | 96 ± 3 |
| | | | overall | 7 ± 5 | 12 ± 6 | 31 ± 2 | 84 ± 15 | 0 ± 0 | 13 ± 6 | 96 ± 2 |
| | | pointmaze-giant-stitch-v0 | task1 | 0 ± 0 | 0 ± 0 | 0 ± 0 | 99 ± 2 | 0 ± 0 | 0 ± 0 | 100 ± 1 |
| | | | task2 | 0 ± 0 | 0 ± 0 | 0 ± 0 | 80 ± 27 | 0 ± 0 | 0 ± 0 | 99 ± 1 |
| | | | task3 | 0 ± 0 | 0 ± 0 | 0 ± 0 | 3 ± 5 | 0 ± 0 | 0 ± 0 | 96 ± 4 |
| | | | task4 | 0 ± 0 | 0 ± 0 | 0 ± 0 | 63 ± 23 | 0 ± 0 | 0 ± 0 | 100 ± 0 |
| | | | task5 | 0 ± 0 | 0 ± 0 | 0 ± 0 | 4 ± 8 | 0 ± 0 | 0 ± 0 | 97 ± 2 |
| | | | overall | 0 ± 0 | 0 ± 0 | 0 ± 0 | 50 ± 8 | 0 ± 0 | 0 ± 0 | 98 ± 1 |
| | | pointmaze-teleport-stitch-v0 | task1 | 28 ± 20 | 34 ± 14 | 0 ± 0 | 0 ± 0 | 0 ± 0 | 24 ± 13 | 53 ± 6 |
| | | | task2 | 13 ± 15 | 41 ± 8 | 12 ± 14 | 7 ± 7 | 0 ± 0 | 23 ± 11 | 4 ± 5 |
| | | | task3 | 48 ± 8 | 50 ± 5 | 47 ± 2 | 15 ± 13 | 0 ± 0 | 46 ± 9 | 0 ± 0 |
| | | | task4 | 40 ± 16 | 50 ± 6 | 46 ± 5 | 19 ± 12 | 8 ± 8 | 46 ± 5 | 5 ± 13 |
| | | | task5 | 29 ± 16 | 48 ± 5 | 21 ± 7 | 1 ± 3 | 13 ± 13 | 31 ± 10 | 10 ± 17 |
| | | | overall | 31 ± 9 | 44 ± 2 | 25 ± 3 | 9 ± 5 | 4 ± 3 | 34 ± 4 | 13 ± 4 |

*Table 12.* Full Results on *antmaze*.

| Environment Type | Dataset Type | Dataset | Task | GCBC | GCIVL | GCIQL | QRL | CRL | HIQL | ALPS |
|---|---|---|---|---|---|---|---|---|---|---|
| antmaze | navigate | antmaze-medium-navigate-v0 | task1 | 35 ± 9 | 81 ± 10 | 63 ± 9 | 93 ± 2 | 97 ± 1 | 94 ± 2 | 97 ± 2 |
| | | | task2 | 21 ± 7 | 85 ± 5 | 78 ± 8 | 90 ± 5 | 95 ± 2 | 97 ± 1 | 98 ± 2 |
| | | | task3 | 28 ± 6 | 60 ± 13 | 71 ± 8 | 86 ± 6 | 92 ± 3 | 96 ± 2 | 98 ± 3 |
| | | | task4 | 28 ± 7 | 42 ± 25 | 59 ± 12 | 83 ± 4 | 94 ± 5 | 96 ± 2 | 97 ± 2 |
| | | | task5 | 37 ± 10 | 92 ± 3 | 85 ± 7 | 88 ± 8 | 96 ± 2 | 96 ± 2 | 96 ± 4 |
| | | | overall | 29 ± 4 | 72 ± 8 | 71 ± 4 | 88 ± 3 | 95 ± 1 | 96 ± 1 | 97 ± 2 |
| | | antmaze-large-navigate-v0 | task1 | 6 ± 3 | 16 ± 12 | 21 ± 6 | 71 ± 15 | 91 ± 3 | 93 ± 3 | 94 ± 3 |
| | | | task2 | 16 ± 4 | 5 ± 6 | 25 ± 7 | 77 ± 7 | 62 ± 14 | 78 ± 9 | 70 ± 39 |
| | | | task3 | 65 ± 4 | 49 ± 18 | 80 ± 5 | 94 ± 2 | 91 ± 2 | 96 ± 2 | 98 ± 2 |
| | | | task4 | 14 ± 3 | 2 ± 2 | 19 ± 6 | 64 ± 8 | 85 ± 11 | 94 ± 2 | 94 ± 4 |
| | | | task5 | 18 ± 4 | 5 ± 2 | 26 ± 9 | 67 ± 9 | 85 ± 3 | 94 ± 3 | 95 ± 4 |
| | | | overall | 24 ± 2 | 16 ± 5 | 34 ± 4 | 75 ± 6 | 83 ± 4 | 91 ± 2 | 93 ± 5 |
| | | antmaze-giant-navigate-v0 | task1 | 0 ± 0 | 0 ± 0 | 0 ± 0 | 1 ± 2 | 2 ± 2 | 47 ± 10 | 62 ± 11 |
| | | | task2 | 0 ± 0 | 0 ± 0 | 0 ± 0 | 17 ± 5 | 21 ± 10 | 74 ± 5 | 64 ± 22 |
| | | | task3 | 0 ± 0 | 0 ± 0 | 0 ± 0 | 14 ± 8 | 5 ± 5 | 55 ± 7 | 77 ± 31 |
| | | | task4 | 0 ± 0 | 0 ± 0 | 0 ± 0 | 18 ± 6 | 35 ± 9 | 69 ± 5 | 84 ± 7 |
| | | | task5 | 1 ± 1 | 1 ± 1 | 1 ± 1 | 18 ± 5 | 16 ± 10 | 82 ± 4 | 58 ± 11 |
| | | | overall | 0 ± 0 | 0 ± 0 | 0 ± 0 | 14 ± 3 | 16 ± 3 | 65 ± 5 | 69 ± 9 |
| | | antmaze-teleport-navigate-v0 | task1 | 17 ± 5 | 35 ± 5 | 26 ± 5 | 31 ± 6 | 35 ± 5 | 37 ± 5 | 33 ± 15 |
| | | | task2 | 51 ± 5 | 41 ± 5 | 58 ± 8 | 47 ± 22 | 92 ± 3 | 66 ± 8 | 52 ± 7 |
| | | | task3 | 22 ± 3 | 36 ± 8 | 31 ± 5 | 35 ± 6 | 47 ± 4 | 37 ± 5 | 43 ± 18 |
| | | | task4 | 25 ± 5 | 45 ± 3 | 33 ± 5 | 33 ± 6 | 50 ± 2 | 30 ± 2 | 46 ± 8 |
| | | | task5 | 14 ± 6 | 38 ± 6 | 26 ± 9 | 28 ± 8 | 44 ± 3 | 41 ± 8 | 40 ± 8 |
| | | | overall | 26 ± 3 | 39 ± 3 | 35 ± 5 | 35 ± 5 | 53 ± 2 | 42 ± 3 | 45 ± 3 |
| | stitch | antmaze-medium-stitch-v0 | task1 | 70 ± 33 | 76 ± 13 | 17 ± 12 | 43 ± 20 | 43 ± 10 | 92 ± 2 | 88 ± 13 |
| | | | task2 | 65 ± 19 | 80 ± 4 | 22 ± 16 | 61 ± 12 | 46 ± 14 | 94 ± 3 | 91 ± 12 |
| | | | task3 | 21 ± 15 | 16 ± 12 | 41 ± 9 | 72 ± 29 | 46 ± 17 | 95 ± 2 | 98 ± 2 |
| | | | task4 | 1 ± 2 | 0 ± 0 | 32 ± 9 | 80 ± 9 | 53 ± 19 | 93 ± 2 | 98 ± 1 |
| | | | task5 | 70 ± 33 | 47 ± 20 | 34 ± 14 | 41 ± 18 | 75 ± 8 | 95 ± 3 | 98 ± 3 |
| | | | overall | 45 ± 11 | 44 ± 6 | 29 ± 6 | 59 ± 7 | 53 ± 6 | 94 ± 1 | 93 ± 7 |
| | | antmaze-large-stitch-v0 | task1 | 2 ± 2 | 23 ± 9 | 0 ± 0 | 7 ± 5 | 1 ± 1 | 85 ± 5 | 95 ± 2 |
| | | | task2 | 0 ± 0 | 0 ± 0 | 0 ± 0 | 10 ± 5 | 4 ± 4 | 24 ± 16 | 94 ± 4 |
| | | | task3 | 15 ± 14 | 69 ± 6 | 37 ± 10 | 73 ± 8 | 43 ± 11 | 94 ± 3 | 99 ± 2 |
| | | | task4 | 0 ± 0 | 0 ± 0 | 0 ± 0 | 1 ± 1 | 5 ± 5 | 70 ± 8 | 95 ± 3 |
| | | | task5 | 0 ± 0 | 0 ± 0 | 0 ± 0 | 1 ± 1 | 1 ± 2 | 60 ± 9 | 93 ± 4 |
| | | | overall | 3 ± 3 | 18 ± 2 | 7 ± 2 | 18 ± 2 | 11 ± 2 | 67 ± 5 | 95 ± 2 |
| | | antmaze-giant-stitch-v0 | task1 | 0 ± 0 | 0 ± 0 | 0 ± 0 | 0 ± 0 | 0 ± 0 | 0 ± 1 | 87 ± 5 |
| | | | task2 | 0 ± 0 | 0 ± 0 | 0 ± 0 | 0 ± 0 | 0 ± 0 | 5 ± 5 | 94 ± 4 |
| | | | task3 | 0 ± 0 | 0 ± 0 | 0 ± 0 | 0 ± 0 | 0 ± 0 | 0 ± 0 | 89 ± 6 |
| | | | task4 | 0 ± 0 | 0 ± 0 | 0 ± 0 | 0 ± 0 | 0 ± 0 | 3 ± 3 | 88 ± 9 |
| | | | task5 | 0 ± 0 | 0 ± 0 | 0 ± 0 | 2 ± 2 | 0 ± 0 | 0 ± 1 | 96 ± 4 |
| | | | overall | 0 ± 0 | 0 ± 0 | 0 ± 0 | 0 ± 0 | 0 ± 0 | 2 ± 2 | 92 ± 3 |
| | | antmaze-teleport-stitch-v0 | task1 | 21 ± 13 | 39 ± 7 | 12 ± 4 | 22 ± 6 | 30 ± 6 | 44 ± 5 | 43 ± 7 |
| | | | task2 | 39 ± 12 | 44 ± 6 | 18 ± 7 | 22 ± 6 | 30 ± 4 | 42 ± 3 | 26 ± 17 |
| | | | task3 | 34 ± 12 | 36 ± 8 | 18 ± 4 | 25 ± 7 | 23 ± 11 | 26 ± 4 | 29 ± 22 |
| | | | task4 | 46 ± 6 | 44 ± 4 | 18 ± 5 | 24 ± 9 | 38 ± 4 | 26 ± 4 | 13 ± 21 |
| | | | task5 | 16 ± 14 | 33 ± 6 | 17 ± 6 | 26 ± 5 | 32 ± 7 | 40 ± 6 | 40 ± 11 |
| | | | overall | 31 ± 6 | 39 ± 3 | 17 ± 2 | 24 ± 5 | 31 ± 4 | 36 ± 2 | 35 ± 11 |
| | explore | antmaze-medium-explore-v0 | task1 | 3 ± 6 | 10 ± 8 | 12 ± 6 | 1 ± 1 | 2 ± 2 | 29 ± 17 | 99 ± 1 |
| | | | task2 | 1 ± 2 | 74 ± 9 | 53 ± 8 | 1 ± 1 | 8 ± 6 | 84 ± 10 | 100 ± 1 |
| | | | task3 | 1 ± 2 | 0 ± 0 | 0 ± 0 | 3 ± 5 | 4 ± 6 | 18 ± 24 | 100 ± 1 |
| | | | task4 | 0 ± 0 | 0 ± 0 | 0 ± 0 | 0 ± 0 | 0 ± 0 | 0 ± 0 | 100 ± 1 |
| | | | task5 | 3 ± 4 | 10 ± 6 | 0 ± 0 | 1 ± 1 | 2 ± 2 | 52 ± 27 | 100 ± 0 |
| | | | overall | 2 ± 1 | 19 ± 3 | 13 ± 2 | 1 ± 1 | 3 ± 2 | 37 ± 10 | 100 ± 0 |
| | | antmaze-large-explore-v0 | task1 | 0 ± 0 | 37 ± 12 | 1 ± 1 | 0 ± 0 | 0 ± 1 | 1 ± 3 | 86 ± 35 |
| | | | task2 | 0 ± 0 | 0 ± 0 | 0 ± 0 | 0 ± 0 | 0 ± 0 | 0 ± 0 | 73 ± 45 |
| | | | task3 | 0 ± 0 | 12 ± 6 | 1 ± 1 | 0 ± 0 | 1 ± 1 | 18 ± 24 | 74 ± 46 |
| | | | task4 | 0 ± 0 | 0 ± 0 | 0 ± 0 | 0 ± 0 | 0 ± 0 | 0 ± 0 | 98 ± 2 |
| | | | task5 | 0 ± 0 | 0 ± 0 | 0 ± 0 | 0 ± 0 | 0 ± 0 | 0 ± 0 | 74 ± 46 |
| | | | overall | 0 ± 0 | 10 ± 3 | 0 ± 0 | 0 ± 0 | 0 ± 0 | 4 ± 5 | 90 ± 15 |
| | | antmaze-teleport-explore-v0 | task1 | 2 ± 2 | 32 ± 4 | 0 ± 1 | 0 ± 0 | 2 ± 1 | 32 ± 11 | 42 ± 11 |
| | | | task2 | 0 ± 0 | 2 ± 4 | 8 ± 6 | 0 ± 1 | 5 ± 4 | 33 ± 17 | 48 ± 21 |
| | | | task3 | 4 ± 3 | 48 ± 3 | 13 ± 8 | 4 ± 4 | 47 ± 6 | 34 ± 16 | 50 ± 6 |
| | | | task4 | 2 ± 2 | 47 ± 5 | 14 ± 8 | 4 ± 4 | 16 ± 12 | 37 ± 19 | 52 ± 11 |
| | | | task5 | 4 ± 2 | 31 ± 2 | 2 ± 1 | 3 ± 3 | 28 ± 5 | 34 ± 14 | 42 ± 10 |
| | | | overall | 2 ± 1 | 32 ± 2 | 7 ± 3 | 2 ± 2 | 20 ± 2 | 34 ± 15 | 48 ± 6 |

*Table 13.* Full Results on *humanoidmaze*.

| Environment Type | Dataset Type | Dataset | Task | GCBC | GCIVL | GCIQL | QRL | CRL | HIQL | ALPS |
|---|---|---|---|---|---|---|---|---|---|---|
| humanoidmaze | navigate | humanoidmaze-medium-navigate-v0 | task1 | 4 ± 1 | 22 ± 5 | 23 ± 6 | 12 ± 7 | 84 ± 3 | 95 ± 2 | 86 ± 5 |
| | | | task2 | 8 ± 4 | 42 ± 8 | 49 ± 6 | 25 ± 8 | 80 ± 5 | 96 ± 2 | 91 ± 5 |
| | | | task3 | 12 ± 3 | 15 ± 3 | 12 ± 6 | 25 ± 10 | 43 ± 11 | 79 ± 6 | 90 ± 5 |
| | | | task4 | 2 ± 1 | 0 ± 0 | 1 ± 0 | 16 ± 7 | 5 ± 5 | 75 ± 6 | 78 ± 32 |
| | | | task5 | 12 ± 4 | 40 ± 8 | 51 ± 8 | 29 ± 12 | 87 ± 7 | 97 ± 1 | 93 ± 4 |
| | | | overall | 8 ± 2 | 24 ± 2 | 27 ± 2 | 21 ± 8 | 60 ± 4 | 89 ± 2 | 89 ± 5 |
| | | humanoidmaze-large-navigate-v0 | task1 | 1 ± 1 | 6 ± 2 | 3 ± 2 | 3 ± 2 | 36 ± 11 | 67 ± 4 | 49 ± 13 |
| | | | task2 | 0 ± 0 | 0 ± 0 | 0 ± 0 | 0 ± 0 | 0 ± 0 | 2 ± 3 | 30 ± 20 |
| | | | task3 | 3 ± 1 | 6 ± 2 | 5 ± 2 | 17 ± 6 | 54 ± 17 | 88 ± 3 | 78 ± 7 |
| | | | task4 | 2 ± 1 | 0 ± 0 | 1 ± 1 | 4 ± 2 | 23 ± 11 | 42 ± 11 | 63 ± 8 |
| | | | task5 | 1 ± 1 | 1 ± 1 | 1 ± 1 | 2 ± 1 | 6 ± 4 | 47 ± 10 | 59 ± 10 |
| | | | overall | 1 ± 0 | 2 ± 1 | 2 ± 1 | 5 ± 1 | 24 ± 4 | 49 ± 4 | 56 ± 5 |
| | | humanoidmaze-giant-navigate-v0 | task1 | 0 ± 0 | 0 ± 0 | 0 ± 0 | 0 ± 0 | 1 ± 1 | 13 ± 7 | 72 ± 7 |
| | | | task2 | 0 ± 0 | 1 ± 1 | 1 ± 1 | 2 ± 1 | 9 ± 5 | 35 ± 11 | 56 ± 23 |
| | | | task3 | 0 ± 0 | 0 ± 0 | 0 ± 0 | 0 ± 0 | 2 ± 2 | 11 ± 4 | 57 ± 24 |
| | | | task4 | 0 ± 0 | 0 ± 0 | 0 ± 0 | 0 ± 0 | 3 ± 2 | 2 ± 2 | 75 ± 8 |
| | | | task5 | 1 ± 1 | 0 ± 0 | 1 ± 1 | 2 ± 1 | 1 ± 1 | 2 ± 2 | 84 ± 4 |
| | | | overall | 0 ± 0 | 0 ± 0 | 0 ± 0 | 1 ± 0 | 3 ± 2 | 12 ± 4 | 67 ± 11 |
| | stitch | humanoidmaze-medium-stitch-v0 | task1 | 20 ± 7 | 13 ± 3 | 12 ± 3 | 6 ± 5 | 27 ± 7 | 84 ± 5 | 53 ± 10 |
| | | | task2 | 49 ± 12 | 7 ± 2 | 8 ± 5 | 13 ± 4 | 37 ± 7 | 94 ± 2 | 65 ± 14 |
| | | | task3 | 24 ± 8 | 25 ± 3 | 20 ± 7 | 30 ± 6 | 40 ± 4 | 86 ± 4 | 89 ± 5 |
| | | | task4 | 3 ± 2 | 1 ± 1 | 2 ± 2 | 18 ± 5 | 28 ± 7 | 86 ± 4 | 76 ± 7 |
| | | | task5 | 49 ± 8 | 16 ± 3 | 18 ± 7 | 22 ± 2 | 49 ± 5 | 90 ± 4 | 69 ± 10 |
| | | | overall | 29 ± 5 | 12 ± 2 | 12 ± 3 | 18 ± 2 | 36 ± 2 | 88 ± 2 | 68 ± 5 |
| | | humanoidmaze-large-stitch-v0 | task1 | 3 ± 4 | 2 ± 1 | 1 ± 1 | 0 ± 0 | 0 ± 0 | 21 ± 5 | 29 ± 12 |
| | | | task2 | 0 ± 0 | 0 ± 0 | 0 ± 0 | 0 ± 0 | 0 ± 0 | 5 ± 2 | 11 ± 9 |
| | | | task3 | 20 ± 11 | 3 ± 2 | 1 ± 1 | 16 ± 7 | 13 ± 3 | 84 ± 4 | 79 ± 7 |
| | | | task4 | 2 ± 1 | 1 ± 1 | 0 ± 1 | 1 ± 1 | 4 ± 1 | 19 ± 4 | 46 ± 15 |
| | | | task5 | 2 ± 2 | 1 ± 1 | 0 ± 0 | 0 ± 0 | 3 ± 1 | 12 ± 2 | 39 ± 14 |
| | | | overall | 6 ± 3 | 1 ± 1 | 0 ± 0 | 3 ± 1 | 4 ± 1 | 28 ± 3 | 39 ± 6 |
| | | humanoidmaze-giant-stitch-v0 | task1 | 0 ± 0 | 0 ± 0 | 0 ± 0 | 0 ± 0 | 0 ± 0 | 1 ± 2 | 59 ± 13 |
| | | | task2 | 0 ± 0 | 1 ± 1 | 0 ± 0 | 1 ± 1 | 0 ± 0 | 12 ± 6 | 63 ± 10 |
| | | | task3 | 0 ± 0 | 0 ± 0 | 0 ± 0 | 0 ± 0 | 0 ± 0 | 2 ± 2 | 49 ± 14 |
| | | | task4 | 0 ± 0 | 0 ± 0 | 0 ± 0 | 0 ± 0 | 0 ± 0 | 1 ± 1 | 72 ± 10 |
| | | | task5 | 0 ± 0 | 0 ± 0 | 1 ± 1 | 1 ± 1 | 0 ± 1 | 0 ± 1 | 84 ± 9 |
| | | | overall | 0 ± 0 | 0 ± 0 | 0 ± 0 | 0 ± 0 | 0 ± 0 | 3 ± 2 | 62 ± 6 |

*Table 14.* Full Results on *cube*.

| Environment Type | Dataset Type | Dataset | Task | GCBC | GCIVL | GCIQL | QRL | CRL | HIQL | ALPS |
|---|---|---|---|---|---|---|---|---|---|---|
| cube | play | cube-single-play-v0 | task1 | 7 ± 3 | 57 ± 6 | 71 ± 9 | 6 ± 2 | 20 ± 6 | 15 ± 5 | 84 ± 11 |
| | | | task2 | 5 ± 2 | 51 ± 6 | 71 ± 6 | 5 ± 2 | 20 ± 4 | 16 ± 5 | 70 ± 9 |
| | | | task3 | 7 ± 3 | 55 ± 6 | 70 ± 6 | 4 ± 1 | 21 ± 6 | 16 ± 3 | 78 ± 9 |
| | | | task4 | 4 ± 2 | 50 ± 4 | 61 ± 8 | 4 ± 2 | 16 ± 3 | 14 ± 5 | 58 ± 14 |
| | | | task5 | 4 ± 2 | 52 ± 6 | 67 ± 7 | 4 ± 3 | 15 ± 3 | 13 ± 4 | 51 ± 13 |
| | | | overall | 6 ± 2 | 53 ± 4 | 68 ± 6 | 5 ± 1 | 19 ± 2 | 15 ± 2 | 68 ± 6 |
| | | cube-double-play-v0 | task1 | 6 ± 3 | 58 ± 5 | 74 ± 8 | 6 ± 3 | 30 ± 7 | 22 ± 6 | 9 ± 6 |
| | | | task2 | 0 ± 0 | 51 ± 6 | 55 ± 11 | 0 ± 0 | 9 ± 2 | 4 ± 3 | 0 ± 1 |
| | | | task3 | 0 ± 0 | 42 ± 7 | 45 ± 7 | 0 ± 0 | 6 ± 1 | 3 ± 2 | 0 ± 0 |
| | | | task4 | 0 ± 0 | 7 ± 2 | 4 ± 3 | 0 ± 0 | 0 ± 0 | 1 ± 1 | 0 ± 0 |
| | | | task5 | 0 ± 0 | 21 ± 1 | 23 ± 6 | 0 ± 0 | 10 ± 2 | 6 ± 2 | 1 ± 1 |
| | | | overall | 1 ± 1 | 36 ± 3 | 40 ± 5 | 1 ± 0 | 10 ± 2 | 6 ± 2 | 2 ± 1 |

*Table 15.* Full Results on *scene*.

| Environment Type | Dataset Type | Dataset | Task | GCBC | GCIVL | GCIQL | QRL | CRL | HIQL | ALPS |
|---|---|---|---|---|---|---|---|---|---|---|
| scene | play | scene-play-v0 | task1 | 18 ± 7 | 75 ± 5 | 93 ± 4 | 19 ± 4 | 49 ± 7 | 40 ± 4 | 90 ± 6 |
| | | | task2 | 1 ± 1 | 62 ± 8 | 82 ± 8 | 1 ± 1 | 12 ± 4 | 40 ± 5 | 4 ± 4 |
| | | | task3 | 2 ± 1 | 64 ± 7 | 72 ± 10 | 1 ± 1 | 26 ± 8 | 36 ± 5 | 29 ± 13 |
| | | | task4 | 3 ± 2 | 7 ± 4 | 8 ± 3 | 5 ± 2 | 5 ± 2 | 55 ± 5 | 2 ± 3 |
| | | | task5 | 0 ± 0 | 2 ± 1 | 1 ± 1 | 0 ± 1 | 1 ± 1 | 20 ± 5 | 0 ± 1 |
| | | | overall | 5 ± 1 | 42 ± 4 | 51 ± 4 | 5 ± 1 | 19 ± 2 | 38 ± 3 | 26 ± 3 |

*Table 16.* Full Results on *visual-antmaze*.

| Environment Type | Dataset Type | Dataset | Task | GCBC | GCIVL | GCIQL | QRL | CRL | HIQL | ALPS |
|---|---|---|---|---|---|---|---|---|---|---|
| visual-antmaze | navigate | visual-antmaze-medium-navigate-v0 | task1 | 17 ± 6 | 30 ± 7 | 16 ± 3 | 0 ± 0 | 92 ± 2 | 90 ± 4 | 92 ± 4 |
| | | | task2 | 8 ± 2 | 21 ± 6 | 7 ± 2 | 0 ± 0 | 94 ± 2 | 92 ± 7 | 96 ± 3 |
| | | | task3 | 17 ± 1 | 24 ± 5 | 16 ± 4 | 0 ± 0 | 98 ± 1 | 94 ± 4 | 98 ± 2 |
| | | | task4 | 12 ± 2 | 21 ± 3 | 9 ± 2 | 0 ± 0 | 94 ± 2 | 94 ± 2 | 98 ± 2 |
| | | | task5 | 4 ± 2 | 16 ± 5 | 6 ± 2 | 0 ± 0 | 94 ± 2 | 94 ± 5 | 94 ± 6 |
| | | | overall | 11 ± 2 | 22 ± 2 | 11 ± 1 | 0 ± 0 | 94 ± 1 | 93 ± 4 | 94 ± 5 |
| | | visual-antmaze-large-navigate-v0 | task1 | 3 ± 1 | 7 ± 2 | 4 ± 3 | 0 ± 0 | 78 ± 5 | 60 ± 10 | 83 ± 3 |
| | | | task2 | 4 ± 3 | 4 ± 1 | 2 ± 1 | 0 ± 0 | 80 ± 3 | 28 ± 9 | 85 ± 6 |
| | | | task3 | 4 ± 2 | 6 ± 2 | 4 ± 1 | 1 ± 1 | 90 ± 3 | 85 ± 10 | 96 ± 4 |
| | | | task4 | 4 ± 2 | 5 ± 3 | 6 ± 1 | 0 ± 1 | 88 ± 3 | 46 ± 7 | 95 ± 3 |
| | | | task5 | 4 ± 2 | 5 ± 1 | 4 ± 2 | 0 ± 0 | 83 ± 2 | 44 ± 10 | 90 ± 4 |
| | | | overall | 4 ± 0 | 5 ± 1 | 4 ± 1 | 0 ± 0 | 84 ± 1 | 53 ± 9 | 88 ± 3 |
| | | visual-antmaze-giant-navigate-v0 | task1 | 0 ± 0 | 0 ± 0 | 0 ± 0 | 0 ± 0 | 17 ± 2 | 2 ± 1 | 9 ± 3 |
| | | | task2 | 1 ± 1 | 2 ± 1 | 1 ± 1 | 0 ± 0 | 73 ± 9 | 12 ± 8 | 30 ± 18 |
| | | | task3 | 0 ± 0 | 0 ± 0 | 0 ± 0 | 0 ± 0 | 22 ± 6 | 2 ± 3 | 21 ± 10 |
| | | | task4 | 0 ± 1 | 0 ± 1 | 0 ± 0 | 0 ± 0 | 47 ± 5 | 4 ± 2 | 48 ± 12 |
| | | | task5 | 1 ± 1 | 2 ± 3 | 1 ± 0 | 0 ± 1 | 77 ± 5 | 13 ± 11 | 58 ± 11 |
| | | | overall | 1 ± 1 | 1 ± 1 | 0 ± 0 | 0 ± 0 | 47 ± 2 | 6 ± 4 | 36 ± 7 |
| | | visual-antmaze-teleport-navigate-v0 | task1 | 2 ± 2 | 6 ± 1 | 2 ± 1 | 3 ± 2 | 32 ± 3 | 32 ± 5 | 44 ± 5 |
| | | | task2 | 6 ± 3 | 9 ± 3 | 9 ± 2 | 6 ± 4 | 73 ± 8 | 40 ± 6 | 46 ± 5 |
| | | | task3 | 9 ± 1 | 12 ± 3 | 9 ± 2 | 10 ± 4 | 47 ± 3 | 33 ± 1 | 46 ± 4 |
| | | | task4 | 10 ± 2 | 10 ± 2 | 8 ± 3 | 6 ± 4 | 50 ± 4 | 44 ± 5 | 42 ± 4 |
| | | | task5 | 1 ± 1 | 3 ± 1 | 3 ± 1 | 4 ± 2 | 36 ± 5 | 33 ± 7 | 47 ± 3 |
| | | | overall | 5 ± 1 | 8 ± 1 | 6 ± 1 | 6 ± 3 | 48 ± 2 | 37 ± 2 | 47 ± 3 |
| | stitch | visual-antmaze-medium-stitch-v0 | task1 | 80 ± 4 | 0 ± 1 | 0 ± 0 | 0 ± 0 | 33 ± 4 | 75 ± 8 | 89 ± 7 |
| | | | task2 | 90 ± 4 | 1 ± 2 | 0 ± 0 | 0 ± 0 | 69 ± 5 | 85 ± 7 | 92 ± 4 |
| | | | task3 | 69 ± 18 | 15 ± 6 | 8 ± 1 | 0 ± 0 | 88 ± 1 | 92 ± 1 | 98 ± 2 |
| | | | task4 | 1 ± 1 | 7 ± 4 | 3 ± 1 | 0 ± 1 | 70 ± 12 | 88 ± 4 | 95 ± 1 |
| | | | task5 | 97 ± 1 | 6 ± 3 | 1 ± 1 | 0 ± 0 | 85 ± 5 | 93 ± 1 | 97 ± 1 |
| | | | overall | 67 ± 4 | 6 ± 2 | 2 ± 0 | 0 ± 0 | 69 ± 2 | 87 ± 2 | 95 ± 2 |
| | | visual-antmaze-large-stitch-v0 | task1 | 26 ± 11 | 0 ± 0 | 0 ± 0 | 0 ± 0 | 6 ± 1 | 36 ± 5 | 88 ± 5 |
| | | | task2 | 0 ± 0 | 0 ± 0 | 0 ± 0 | 0 ± 0 | 2 ± 1 | 3 ± 2 | 84 ± 4 |
| | | | task3 | 73 ± 14 | 3 ± 2 | 0 ± 0 | 2 ± 2 | 36 ± 10 | 87 ± 6 | 98 ± 2 |
| | | | task4 | 7 ± 5 | 1 ± 1 | 0 ± 0 | 1 ± 1 | 8 ± 1 | 7 ± 4 | 94 ± 2 |
| | | | task5 | 11 ± 5 | 0 ± 0 | 0 ± 0 | 0 ± 0 | 5 ± 2 | 6 ± 1 | 92 ± 6 |
| | | | overall | 24 ± 3 | 1 ± 1 | 0 ± 0 | 1 ± 1 | 11 ± 3 | 28 ± 2 | 90 ± 2 |
| | | visual-antmaze-giant-stitch-v0 | task1 | 0 ± 0 | 0 ± 0 | 0 ± 0 | 0 ± 0 | 0 ± 0 | 0 ± 0 | 24 ± 4 |
| | | | task2 | 1 ± 2 | 0 ± 0 | 0 ± 0 | 0 ± 0 | 0 ± 0 | 1 ± 1 | 74 ± 8 |
| | | | task3 | 0 ± 0 | 0 ± 0 | 0 ± 0 | 0 ± 0 | 0 ± 0 | 0 ± 0 | 40 ± 19 |
| | | | task4 | 0 ± 0 | 0 ± 0 | 0 ± 0 | 0 ± 0 | 0 ± 0 | 0 ± 0 | 59 ± 6 |
| | | | task5 | 0 ± 0 | 0 ± 0 | 0 ± 0 | 0 ± 0 | 0 ± 1 | 0 ± 0 | 88 ± 8 |
| | | | overall | 0 ± 0 | 0 ± 0 | 0 ± 0 | 0 ± 0 | 0 ± 0 | 0 ± 0 | 55 ± 6 |
| | | visual-antmaze-teleport-stitch-v0 | task1 | 37 ± 4 | 2 ± 2 | 1 ± 1 | 0 ± 0 | 20 ± 5 | 36 ± 5 | 48 ± 6 |
| | | | task2 | 36 ± 3 | 2 ± 1 | 1 ± 1 | 1 ± 1 | 40 ± 9 | 38 ± 3 | 16 ± 12 |
| | | | task3 | 17 ± 6 | 2 ± 1 | 2 ± 1 | 3 ± 4 | 32 ± 9 | 36 ± 5 | 0 ± 0 |
| | | | task4 | 39 ± 9 | 1 ± 1 | 0 ± 0 | 2 ± 3 | 45 ± 7 | 37 ± 6 | 0 ± 0 |
| | | | task5 | 29 ± 1 | 1 ± 1 | 1 ± 1 | 1 ± 1 | 22 ± 9 | 38 ± 5 | 45 ± 8 |
| | | | overall | 32 ± 3 | 1 ± 1 | 1 ± 0 | 1 ± 2 | 32 ± 6 | 37 ± 4 | 21 ± 4 |
| | explore | visual-antmaze-medium-explore-v0 | task1 | 0 ± 0 | 0 ± 0 | 0 ± 0 | 0 ± 0 | 0 ± 0 | 0 ± 0 | 52 ± 20 |
| | | | task2 | 0 ± 0 | 0 ± 0 | 0 ± 0 | 0 ± 0 | 0 ± 0 | 0 ± 0 | 82 ± 27 |
| | | | task3 | 0 ± 0 | 0 ± 0 | 0 ± 0 | 0 ± 0 | 0 ± 0 | 1 ± 2 | 82 ± 6 |
| | | | task4 | 0 ± 0 | 0 ± 0 | 0 ± 0 | 0 ± 0 | 0 ± 0 | 0 ± 0 | 78 ± 20 |
| | | | task5 | 0 ± 0 | 0 ± 0 | 0 ± 0 | 0 ± 0 | 0 ± 0 | 0 ± 0 | 55 ± 23 |
| | | | overall | 0 ± 0 | 0 ± 0 | 0 ± 0 | 0 ± 0 | 0 ± 0 | 0 ± 0 | 78 ± 10 |
| | | visual-antmaze-large-explore-v0 | task1 | 0 ± 0 | 0 ± 0 | 0 ± 0 | 0 ± 0 | 0 ± 0 | 0 ± 0 | 22 ± 7 |
| | | | task2 | 0 ± 0 | 0 ± 0 | 0 ± 0 | 0 ± 0 | 0 ± 0 | 0 ± 0 | 2 ± 2 |
| | | | task3 | 0 ± 0 | 0 ± 0 | 0 ± 0 | 0 ± 0 | 0 ± 0 | 0 ± 0 | 28 ± 26 |
| | | | task4 | 0 ± 0 | 0 ± 0 | 0 ± 0 | 0 ± 0 | 0 ± 0 | 0 ± 0 | 32 ± 23 |
| | | | task5 | 0 ± 0 | 0 ± 0 | 0 ± 0 | 0 ± 0 | 0 ± 0 | 0 ± 0 | 12 ± 12 |
| | | | overall | 0 ± 0 | 0 ± 0 | 0 ± 0 | 0 ± 0 | 0 ± 0 | 0 ± 0 | 26 ± 10 |
| | | visual-antmaze-teleport-explore-v0 | task1 | 0 ± 0 | 0 ± 0 | 0 ± 0 | 0 ± 0 | 0 ± 0 | 0 ± 0 | 6 ± 12 |
| | | | task2 | 0 ± 0 | 0 ± 0 | 0 ± 0 | 0 ± 0 | 0 ± 0 | 0 ± 0 | 31 ± 11 |
| | | | task3 | 0 ± 0 | 0 ± 1 | 0 ± 1 | 0 ± 0 | 3 ± 1 | 38 ± 8 | 48 ± 8 |
| | | | task4 | 0 ± 0 | 0 ± 0 | 0 ± 0 | 0 ± 0 | 0 ± 0 | 28 ± 13 | 41 ± 9 |
| | | | task5 | 0 ± 0 | 0 ± 0 | 0 ± 0 | 0 ± 0 | 2 ± 2 | 27 ± 18 | 32 ± 8 |
| | | | overall | 0 ± 0 | 0 ± 0 | 0 ± 0 | 0 ± 0 | 1 ± 0 | 19 ± 8 | 34 ± 4 |

*Table 17.* **Success rates (%) for the six variants of ALPS compared in Section 6 on the state-based OGBench locomotion tasks.** Results averaged over 8 seeds with standard deviation reported after ±. CEM, $\pi_{\text{prior}}$, and CEM + $\pi_{\text{prior}}$ are the low-level planning only variants. Dijkstra + CEM and Dijkstra + $\pi_{\text{prior}}$ are the hierarchical planning variants with either CEM or $\pi_{\text{prior}}$ acting as the low-level planner.

| Environment | Dataset | CEM | $\pi_{\text{prior}}$ | CEM + $\pi_{\text{prior}}$ | Dijkstra + CEM | Dijkstra + $\pi_{\text{prior}}$ | ALPS |
|---|---|---|---|---|---|---|---|
| pointmaze | pointmaze-medium-navigate-v0 | $58 \pm 8$ | $19 \pm 8$ | $30 \pm 7$ | $100 \pm 0$ | $46 \pm 10$ | $82 \pm 10$ |
| | pointmaze-large-navigate-v0 | $19 \pm 11$ | $17 \pm 10$ | $21 \pm 9$ | $100 \pm 0$ | $44 \pm 17$ | $80 \pm 8$ |
| | pointmaze-giant-navigate-v0 | $0 \pm 0$ | $0 \pm 0$ | $0 \pm 0$ | $52 \pm 10$ | $64 \pm 21$ | $67 \pm 11$ |
| | pointmaze-medium-stitch-v0 | $40 \pm 13$ | $40 \pm 12$ | $46 \pm 11$ | $99 \pm 3$ | $82 \pm 10$ | $94 \pm 6$ |
| | pointmaze-large-stitch-v0 | $0 \pm 0$ | $12 \pm 9$ | $9 \pm 9$ | $100 \pm 1$ | $93 \pm 4$ | $96 \pm 2$ |
| | pointmaze-giant-stitch-v0 | $0 \pm 0$ | $0 \pm 0$ | $0 \pm 0$ | $68 \pm 8$ | $98 \pm 1$ | $98 \pm 1$ |
| | pointmaze-teleport-navigate-v0 | $5 \pm 5$ | $16 \pm 6$ | $20 \pm 6$ | $15 \pm 8$ | $33 \pm 6$ | $40 \pm 6$ |
| | pointmaze-teleport-stitch-v0 | $5 \pm 6$ | $8 \pm 5$ | $9 \pm 6$ | $10 \pm 10$ | $19 \pm 7$ | $13 \pm 4$ |
| antmaze | antmaze-medium-navigate-v0 | $0 \pm 0$ | $50 \pm 10$ | $57 \pm 8$ | $0 \pm 0$ | $92 \pm 5$ | $97 \pm 2$ |
| | antmaze-large-navigate-v0 | $0 \pm 0$ | $30 \pm 11$ | $29 \pm 12$ | $0 \pm 0$ | $90 \pm 4$ | $93 \pm 5$ |
| | antmaze-giant-navigate-v0 | $0 \pm 0$ | $2 \pm 1$ | $2 \pm 2$ | $0 \pm 0$ | $54 \pm 5$ | $69 \pm 9$ |
| | antmaze-medium-stitch-v0 | $0 \pm 0$ | $50 \pm 13$ | $55 \pm 12$ | $0 \pm 0$ | $92 \pm 4$ | $93 \pm 7$ |
| | antmaze-large-stitch-v0 | $0 \pm 0$ | $14 \pm 8$ | $11 \pm 7$ | $0 \pm 0$ | $93 \pm 2$ | $95 \pm 2$ |
| | antmaze-giant-stitch-v0 | $0 \pm 0$ | $0 \pm 0$ | $0 \pm 0$ | $0 \pm 0$ | $88 \pm 3$ | $92 \pm 3$ |
| | antmaze-medium-explore-v0 | $0 \pm 0$ | $11 \pm 6$ | $7 \pm 3$ | $47 \pm 22$ | $92 \pm 4$ | $100 \pm 0$ |
| | antmaze-large-explore-v0 | $0 \pm 0$ | $1 \pm 1$ | $0 \pm 0$ | $11 \pm 8$ | $43 \pm 9$ | $90 \pm 15$ |
| | antmaze-teleport-navigate-v0 | $0 \pm 0$ | $35 \pm 6$ | $38 \pm 6$ | $0 \pm 0$ | $44 \pm 5$ | $45 \pm 3$ |
| | antmaze-teleport-stitch-v0 | $0 \pm 0$ | $21 \pm 8$ | $24 \pm 8$ | $0 \pm 0$ | $33 \pm 10$ | $35 \pm 11$ |
| | antmaze-teleport-explore-v0 | $6 \pm 3$ | $4 \pm 3$ | $9 \pm 5$ | $13 \pm 5$ | $38 \pm 5$ | $48 \pm 6$ |
| humanoidmaze | humanoidmaze-medium-navigate-v0 | $0 \pm 0$ | $29 \pm 12$ | $26 \pm 10$ | $0 \pm 0$ | $86 \pm 6$ | $89 \pm 5$ |
| | humanoidmaze-large-navigate-v0 | $0 \pm 0$ | $3 \pm 2$ | $3 \pm 2$ | $0 \pm 0$ | $58 \pm 6$ | $56 \pm 5$ |
| | humanoidmaze-giant-navigate-v0 | $0 \pm 0$ | $0 \pm 0$ | $0 \pm 0$ | $0 \pm 0$ | $66 \pm 10$ | $67 \pm 11$ |
| | humanoidmaze-medium-stitch-v0 | $0 \pm 0$ | $33 \pm 8$ | $31 \pm 7$ | $0 \pm 0$ | $64 \pm 6$ | $68 \pm 5$ |
| | humanoidmaze-large-stitch-v0 | $0 \pm 0$ | $2 \pm 2$ | $1 \pm 2$ | $0 \pm 0$ | $48 \pm 5$ | $56 \pm 5$ |
| | humanoidmaze-giant-stitch-v0 | $0 \pm 0$ | $0 \pm 1$ | $0 \pm 1$ | $0 \pm 0$ | $53 \pm 10$ | $62 \pm 6$ |

